# Microtubule plus-ends act as physical signaling hubs to activate RhoA during cytokinesis

**Vikash Verma[1], Thomas J Maresca[1,2]\***

[1]Biology Department, University of Massachusetts, Amherst, United States; [2]Molecular and Cellular Biology Graduate Program, University of Massachusetts, Amherst, United States

**Abstract** Microtubules (MTs) are essential for cleavage furrow positioning during cytokinesis, but the mechanisms by which MT-derived signals spatially define regions of cortical contractility are unresolved. In this study cytokinesis regulators visualized in *Drosophila melanogaster* (*Dm*) cells were found to localize to and track MT plus-ends during cytokinesis. The RhoA GEF Pebble (*Dm* ECT2) did not evidently tip-track, but rather localized rapidly to cortical sites contacted by MT plus-tips, resulting in RhoA activation and enrichment of myosin-regulatory light chain. The MT plus-end localization of centralspindlin was compromised following EB1 depletion, which resulted in a higher incidence of cytokinesis failure. Centralspindlin plus-tip localization depended on the C-terminus and a putative EB1-interaction motif (hxxPTxh) in RacGAP50C. We propose that MT plus-end-associated centralspindlin recruits a cortical pool of *Dm* ECT2 upon physical contact to activate RhoA and to trigger localized contractility.

DOI: https://doi.org/10.7554/eLife.38968.001

**\*For correspondence:**
tmaresca@bio.umass.edu

**Competing interests:** The authors declare that no competing interests exist.

## Introduction

Cell division ends with the formation of an actomyosin contractile ring, positioned midway between the segregated chromosomes. Constriction of the ring generates a cleavage furrow that physically divides the cytosol, a process known as cytokinesis. Establishing and maintaining the position of the cleavage furrow between daughter nuclei is vital for embryonic development, tissue and stem cell maintenance, and preserving ploidy (*Li, 2007*). Furthermore, cytokinesis failure yields tetraploid daughter cells and has been shown to potentiate chromosomal instability, metastatic cellular behaviors, and tumorigenesis in mice (*Fujiwara et al., 2005*; *Ganem et al., 2014*; *Ganem et al., 2007*). Therefore, a cell must ensure positioning, maintenance, and completion of the cleavage furrow with high fidelity.

In animal cells, the spindle apparatus specifies the position of the cleavage furrow during anaphase (*Rappaport, 1961*), and it is widely accepted that the cleavage plane is positioned via microtubule (MT)-dependent mechanisms (*Rappaport, 1996*); however, there is lack of consensus regarding the nature and molecular composition of the MT-derived positioning signals and how they are delivered. While it has been posited in different models that distinct MT populations spatially regulate stimulation or relaxation of cortical contractility, decades of work on the topic have revealed the existence of redundant pathways that contribute to furrow positioning to various degrees in different organisms and developmental stages. Given the central importance of cytokinesis it is not surprising that reliability is improved by incorporating redundancies in the system (*Basant and Glotzer, 2018*). Nonetheless, present understanding of how the population of MTs examined in this study (astral MTs) contributes to the robustness of cleavage furrow positioning in unperturbed mitoses is limited.

A highly conserved and essential regulator of furrow positioning is the centralspindlin complex (*White and Glotzer, 2012*). Centralspindlin is a tetrameric complex consisting of a dimer of the kinesin-6 family member MKLP1 (ZEN-4 in *C. elegans*, and Pavarotti in *Drosophila*; referred to in this study as *Dm* MKLP1) bound to a dimer of the GTPase activating protein MgcRacGAP/Cyk4 (CYK-4 in *C. elegans*, and Tumbleweed/RacGAP50C in *Drosophila* - the term RacGAP50C will be used here). Centralspindlin is a robust MT bundler and following anaphase onset the complex becomes highly enriched in MT overlaps. Interestingly, centralspindlin components have also been observed at the cell cortex and at MT plus-ends post-anaphase (*Breznau et al., 2017*; *Minestrini et al., 2003*; *Nishimura and Yonemura, 2006*; *Vale et al., 2009*). The relative functional contributions of the various pools of centralspindlin to furrow positioning are poorly understood.

Spatio-temporal regulation of cytokinesis is also controlled by a complex interplay of cyclin-dependent kinase 1 (CDK1), Aurora B kinase (ABK), and Polo kinase. High CDK1 activity during mitosis results in phosphorylation of the centralspindlin component MKLP1, which prevents its association with MTs until after anaphase onset as CDK1 activity drops (*Glotzer, 2009*; *Mishima et al., 2004*). Binding and enrichment of MKLP1 on the central spindle is further regulated by ABK activity, which stabilizes the midzone localization of the centralspindlin complex via phosphorylation of MKLP1 (*Douglas et al., 2010*). In many cell types, midzone localization of the ABK-containing chromosomal passenger complex (CPC) is dependent on the plus-end directed motor MKLP2 (*Cesario et al., 2006*; *Gruneberg et al., 2004*; *Kitagawa et al., 2013*; *Nguyen et al., 2014*), while ABK's cortical localization depends on actin binding by the CPC component INCENP (*Landino et al., 2017*; *Landino and Ohi, 2016*). While the role of ABK at the central spindle is well documented, its function at the equatorial cortex has not been extensively examined; although a recent study showed that ABK promotes oligomerization of centralspindlin at the membrane (*Basant and Glotzer, 2018*; *Basant et al., 2015*). Polo kinase also plays a central role in cytokinesis (*Brennan et al., 2007*; *Burkard et al., 2009*; *Carmena et al., 2014*; *Llamazares et al., 1991*; *Petronczki et al., 2008*). Polo kinase phosphorylates the centralspindlin component RacGAP50C (*Ebrahimi et al., 2010*), to recruit the RhoGEF, ECT2, to the midzone (*Burkard et al., 2009*; *Petronczki et al., 2007*; *Somers and Saint, 2003*; *Wolfe et al., 2009*). ECT2 produces RhoA-GTP at the membrane, which promotes cortical contractility via activation of downstream actin and myosin regulatory pathways (*Bement et al., 2005*; *Jordan and Canman, 2012*; *Yüce et al., 2005*). PRC1 (Feo in *Drosophila*) and Kinesin-4 (Klp3A in *Drosophila*) are required for Polo kinase recruitment to the midzone in *Drosophila* and mammalian cells (*D'Avino et al., 2007*; *Neef et al., 2007*). However, Feo depletion does not result in cleavage furrow initiation or ingression defects (*D'Avino et al., 2007*), indicating that midzone-localized Polo is not necessary for furrow positioning. Nevertheless, global Polo kinase activity is essential for cytokinesis as it is required for cleavage furrow initiation (*Brennan et al., 2007*; *Lénárt et al., 2007*; *Petronczki et al., 2007*).

In this study, live-cell TIRF microscopy was applied to dividing *Drosophila melanogaster* (*Dm*) S2 cells to visualize the spatio-temporal dynamics of lynchpin cytokinesis regulators. Live-cell imaging revealed that centralspindlin (*Dm* MKLP1 and RacGAP50C), ABK, and Polo each localize to and track astral MT plus-tips within minutes of anaphase onset before being lost from a majority of polar astral MTs and retained on equatorial astral MTs. Specialized MT plus-tips enriched with centralspindlin were deemed ''c̲ytokinesis s̲ignaling TIPs'', referred to hereafter as CS-TIPs, because they recruited cortical ECT2 and locally activated RhoA.

## Results

### The centralspindlin complex, ABK, and Polo kinase localize to astral MT plus-ends following anaphase onset and become patterned onto equatorial astral MTs over time

It has long been known that the centralspindlin complex and CPC are highly enriched in the midzone during cytokinesis (*Glotzer, 2009*); however, previous studies in *Drosophila* and mammalian cells as well as *Xenopus* embryos have reported MT plus-tip localization of the centralspindlin complex as well as the CPC component ABK (*Breznau et al., 2017*; *Nishimura and Yonemura, 2006*; *Vale et al., 2009*). While there has been significant investigation of midzone populations of cytokinesis regulators, very little attention has been given to MT plus-end localized components. Thus, we

sought to further investigate the spatio-temporal dynamics of the tip-localization properties of centralspindlin and ABK by live-cell TIRF microscopy of dividing *Drosophila* S2 cells expressing fluorescently tagged *Dm* MKLP1, RacGAP50C, and ABK.

To specifically observe the MT plus-end localization properties of these regulators, S2 cells, which are semi-adherent, were seeded on concanavalin A (Con A) coated glass-bottom dishes to adhere and flatten them. We had previously shown that contractile myosin rings assembled normally on Con A (*Ye et al., 2016*), but since such a treatment could interfere with furrow formation, the effects of Con A on cytokinesis was further assessed by overnight time-lapse imaging of S2 cells seeded on Con A. Importantly, we found that 97% of dividing cells began to ingress furrows in the mid-cell region within ~10 min. of anaphase onset. Of these cells, 32% completed cytokinesis normally (as defined by clearly delineated mono-nucleated daughter cells) while 68% of cells regressed their furrows 39.4 ± 7.6 min. after furrow ingression initiated (*Video 1*, n = 28, mean ± SD values are reported). Thus, the Con A concentration used in these experiments negatively impacted the latest stages of cytokinesis (most likely cytokinetic abscission) but did not affect the processes that we aimed to study, namely cleavage furrow establishment and ingression.

After observing that Con A didn't interfere with the early stages of cytokinesis, we began by imaging *Dm* MKLP1. In metaphase, *Dm* MKLP1 diffusely localized throughout the cytosol without evident spindle MT enrichment (*Video 2*); however, within 2–3 min. of anaphase onset, *Dm* MKLP1 localized to and tip-tracked on the plus-ends of astral MTs (*Figure 1A*; *Figure 1—figure supplement 1*; *Video 2*). In agreement with prior observations (*Vale et al., 2009*), *Dm* MKLP1 localized to astral MT plus-tips, both polar and equatorial, before being lost from the polar tips and becoming preferentially patterned onto equatorial tips ~ 10 min. following anaphase onset. We next visualized FP-tagged RacGAP50C since homodimers of MKLP1/ZEN-4 and MgcRacGAP/CYK-4 interact to form a functional centralspindlin complex (*Mishima et al., 2002*; *Pavicic-Kaltenbrunner et al., 2007*). In agreement with a recent report of MgcRacGAP localization to the MT plus-ends in *Xeno-*

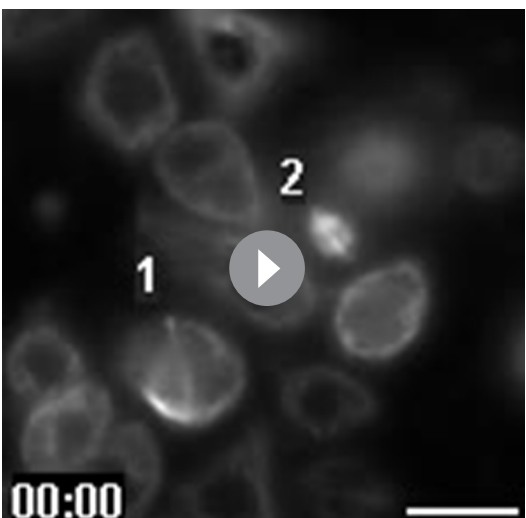

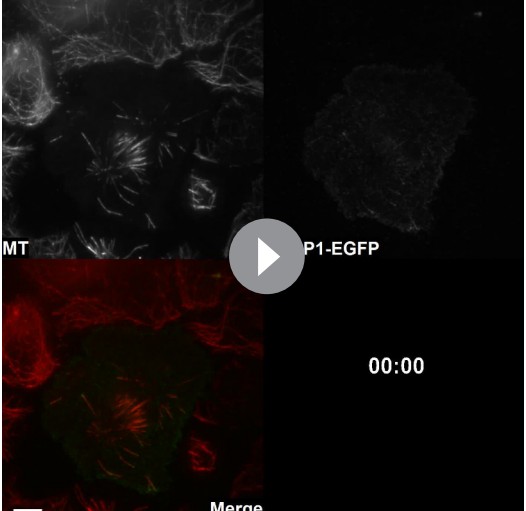

**Video 1.** Con A coated coverslips do not affect early events of cytokinesis. Video shows overnight imaging of *Drosophila* S2 cells on a Con A coated dish, expressing Tag-RFP-T-α-tubulin. There are two diving cells (marked 1 and 2) in this field of view. One cell divided normally (cell number 1), while the other cell ingressed normally but regressed after sometime (cell number 2). Frames were acquired at 4 min. intervals on a Nikon Eclipse Ti-E - TIRF microscope. The playback rate is 10 frames per second. Time: mins:secs. Scale bar, 5 μm.
DOI: https://doi.org/10.7554/eLife.38968.002

**Video 2.** *Dm* MKLP1-EGFP localizes to the CS-TIPs of polar and equatorial MTs. Video shows co-expression of *Dm* MKLP1-EGFP (green) and Tag-RFP-T-α-tubulin (red) in S2 cells. *Dm* MKLP1-EGFP decorates the polar and equatorial CS-TIPs within ~2–3 min. of anaphase onset before becoming patterned onto equatorial MTs (~7–10 min.). *Dm* MKLP1-EGFP is also visible on the growing MT plus-tips and the central spindle. Frames were acquired at 10 s. intervals on a Nikon Eclipse Ti-E - TIRF microscope. The playback rate is 10 frames per second. Time: mins:secs. Scale bar, 5 μm.
DOI: https://doi.org/10.7554/eLife.38968.005

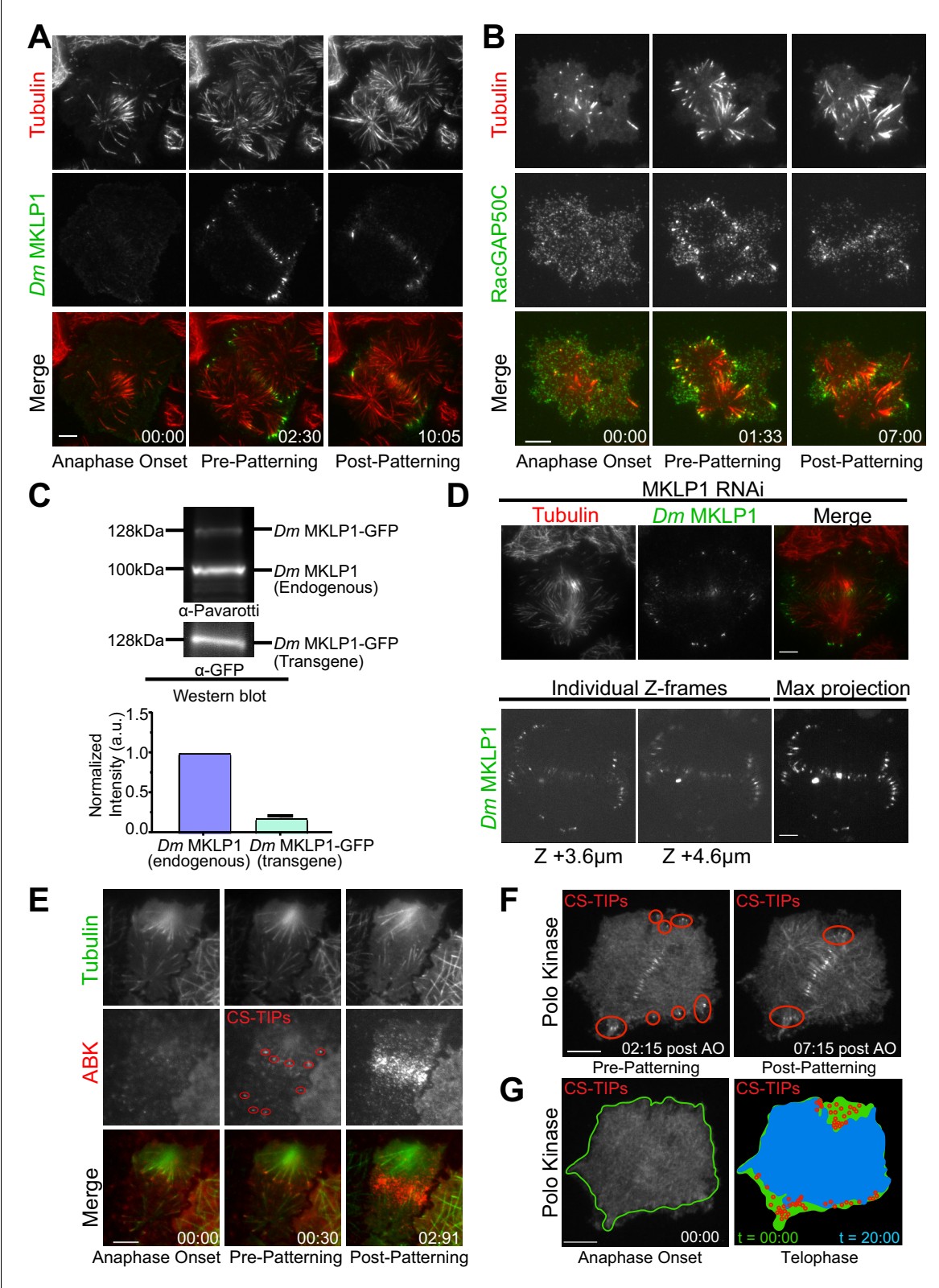

**Figure 1.** *Dm* MKLP1, RacGAP50C, ABK, and polo kinase localize to CS-TIPs and the midzone. (**A**) Selected still frames from the live-cell TIRF movies showing *Dm* MKLP1-EGFP (green) localization to the plus-tips of polar and equatorial MTs (red) within 2.30 min. of anaphase onset. *Dm* MKLP1-EGFP gets patterned on the equatorial MTs after ~10 min. (the last panel). (**B**) Selected still frames from the live-cell TIRF movies showing RacGAP50C-EGFP (green) localization to the plus-tips of polar and equatorial MTs (red) within 1.33 min. of anaphase onset. The last panel shows patterning of

*Figure 1 continued on next page*

*Figure 1 continued*

RacGAP50C-EGFP mainly on the equatorial MTs. (**C**) A representative western blot showing expression of *Dm* MKLP1 transgene and its endogenous counterpart; bar graph shows quantitation of *Dm* MKLP1 intensity from the western blot shown above and other similar blots (bottom). (**D**) Still images from live-cell TIRF microscopy showing *Dm* MKLP1-EGFP localization to the MT plus-tips and midzone in cells where endogenous *Dm* MKLP1 was depleted (top); selected Z-frames from spinning disk confocal z-sections showing *Dm* MKLP1-EGFP localization towards the dorsal surface (up to 4.6 µm) and throughout the cell volume. (**E**) ABK (red) localizes to the plus-tips of polar and equatorial MTs (green) within 30 s. of anaphase onset. The last panel shows ABK localization to the equatorial and midzone MTs. (**F**) A representative image showing Polo localization to the CS-TIPs (red regions) within 02:15 min. of anaphase onset (AO) and then patterning on the equatorial MT tips. (**G**) Images are arranged to simulate a time-series of the events for Polo kinase that occurred during anaphase/telophase. Outline of the cell cortex at anaphase onset (t = 00:00) is marked in green. In the next image, red circles represent the position of CS-TIPs that appeared over the time-series; the blue region indicates the edge of the soluble Polo-EGFP signal after 20:00 minutes; while the green fills the space to the original cell outline (green line). The membrane invaginates where CS-TIPs contact the cortex as evident in the overlay of the cell periphery (blue) relative to the cell boundary at anaphase onset (green). *Figure 1A–D*: zero-time points (00:00) indicate anaphase onset, the next time points indicate the onset of the decoration of cytokinesis regulatory proteins (MKLP1, RagGAP50C, ABK, Polo kinase) on both the polar and equatorial MTs (pre-patterning), and the last time points refer to the time when the vast majority of polar CS-TIPs are lost and are retained on the equatorial MTs (post-patterning). $n \geq 25$ for every condition, at least three independent experiments were performed. Time: mins:secs. Scale bars, 10 µm.

DOI: https://doi.org/10.7554/eLife.38968.003

The following figure supplement is available for figure 1:

**Figure supplement 1.** Depletion phenotypes of *Dm* MKLP1 and RacGAP50C are rescued by expression of their respective transgenes.
DOI: https://doi.org/10.7554/eLife.38968.004

*pus laevis* (*Breznau et al., 2017*), we also observed MT plus-end localization and tip-tracking activity of RacGAP50C during cytokinesis (*Figure 1B*; *Video 3*) Like *Dm* MKLP1, RacGAP50C decorated polar and equatorial astral MT plus-ends within ~2 min. of anaphase onset before becoming patterned onto equatorial MTs. We also observed RacGAP50C and *Dm* MKLP1 accumulation on midzone MTs and the equatorial cortex (*Figure 1A, B*; *Videos 2* and *3*). Thus, the two components of the centralspindlin complex, *Dm* MKLP1 and Rac-GAP50C, exhibit the same dynamic localization pattern on midzone MTs, the cortex, and astral MT plus-tips as cells progressed through ana-phase and cytokinesis.

Since our transgenes are expressed on top of the endogenous proteins, we examined the level of transgene expression to assess whether the appearance of CS-TIPs was a result of overex-pressing centralspindlin components. To address this issue we performed: 1) quantitative western blotting of cells expressing *Dm* MKLP1- and Rac-GAP50C-EGFP to access the extent of overex-pression and 2) TIRF and/or spinning disk confocal imaging under conditions where the endogenous copy of either *Dm* MKLP1 or Rac-GAP50C was depleted using RNAi. Western blot-ting indicated that the *Dm* MKLP1-EGFP transgene was expressed ~5 fold lower than its endogenous counterpart (*Figure 1C*). Impor-tantly, *Dm* MKLP1-EGFP localized to MT plus-ends, midzones, and the cortex when endoge-nous *Dm* MKLP1 was depleted and rescued the severe cytokinesis failure phenotype that resulted from depletion of *Dm* MKLP1. (*Figure 1D*; *Fig-ure 1—figure supplement 1*). Spinning disk con-focal imaging of *Dm* MKLP1-EGFP cells depleted for the endogenous protein revealed that CS-TIPs are not restricted to the ventral surface of

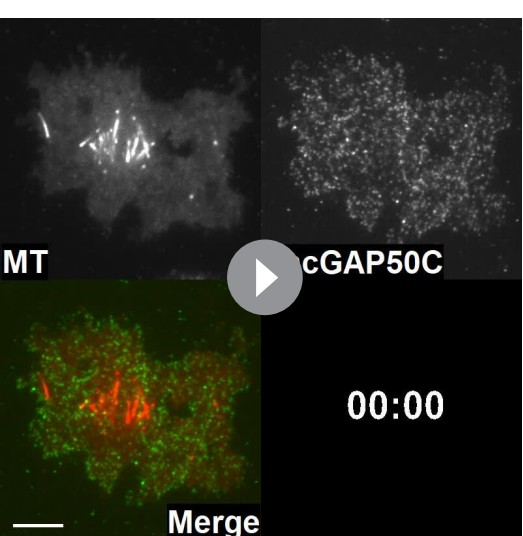

**Video 3.** RacGAP50C-EGFP localizes to the CS-TIPs of polar and equatorial MTs. Video shows co-expression of RacGAP50C-EGFP (green) and Tag-RFP-T-α-tubulin (red) in S2 cells. RacGAP50C-EGFP decorates the CS-TIPs within ~2 min. of anaphase onset before becoming patterned onto equatorial MTs. RacGAP50C-EGFP is also visible on the growing MT plus-tips and the central spindle. Frames were acquired at 5 s. intervals after anaphase onset on a Nikon Eclipse Ti-E - TIRF microscope. The playback rate is 10 frames per second Time: mins:secs. Scale bar, 5 µm.
DOI: https://doi.org/10.7554/eLife.38968.006

the cell as the decoration of CS-TIPs was observed throughout the cell volume (up to 4.6 µm towards the dorsal surface) (*Figure 1D*; *Video 5*). Like *Dm* MKLP1-EGFP, RacGAP50C-EGFP expression was less than its endogenous counterpart (*Figure 1—figure supplement 1*). Expression of Rac-GAP50C-EGFP largely rescued the failure of cytokinesis following RacGAP50C depletion and the RacGAP50C-EGFP localized to CS-TIPs, midzones, and the cortex when the endogenous Rac-GAP50C was depleted (*Figure 1—figure supplement 1*). Altogether, these data lead us to conclude that the spatio-temporal properties of the MT plus-tip population of centralspindlin were not a consequence of transgene overexpression.

Consistent with previous reports (*Landino et al., 2017*; *Vale et al., 2009*), we also observed ABK localization on polar and equatorial astral MT plus-tips, midzones, and the equatorial cortex (*Figure 1E*). Possibly due to intense and rapid ABK enrichment at the cortex, ABK localization on plus-ends was evident for less time (~1 min.) than the centralspindlin components, which typically lasted for 2–10 min. on polar MTs and >10 min. on equatorial MTs. Polo kinase becomes enriched on midzones during cytokinesis and its activity is required for furrow initiation and successful cytokinesis (*Brennan et al., 2007*; *Burkard et al., 2009*; *Carmena et al., 1998*; *Lénárt et al., 2007*; *Petronczki et al., 2007*). We next applied TIRF microscopy to Polo-EGFP expressing cells to examine its spatio-temporal dynamics during cytokinesis. As expected, Polo localized to the overlap zone of the midzone MT array. However, Polo kinase also localized to the MT plus-ends within ~2–3 min. of anaphase onset before becoming patterned onto equatorial astral tips ~10 min. post anaphase onset (*Figure 1F*; *Video 4*). Polo kinase uniformly localized along the length of MTs as cells advanced into late telophase, which was expected since Polo kinase localizes to interphase MTs in *Drosophila* cells (*Archambault et al., 2008*). Thus, prior to telophase, Polo kinase exhibited an identical localization pattern as the centralspindlin complex. While the centralspindlin plus-tip localization is clearly a conserved phenomenon as it has now been observed in *Drosophila*, *Xenopus*, and human cells, it would be worthwhile to determine if the plus-tip localization of Polo kinase is also conserved or rather a *Drosophila*-specific phenomenon

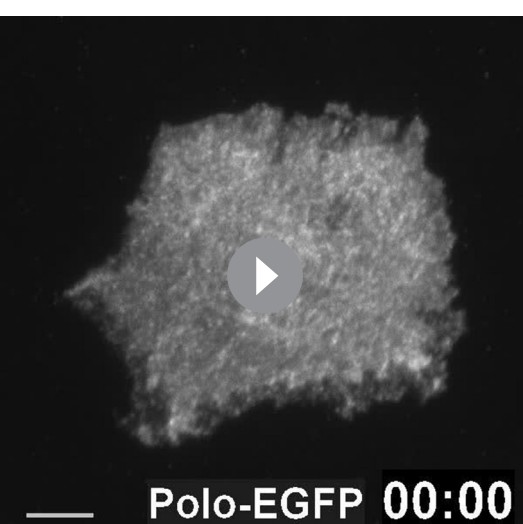

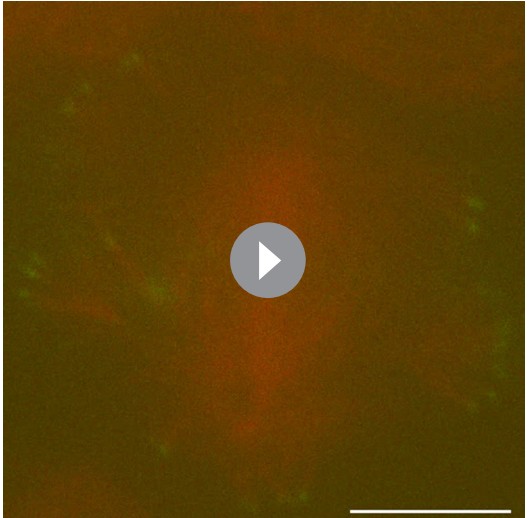

**Video 4.** Polo kinase localizes to the CS-TIPs and triggers cortical contractility. Video shows the expression of Polo-EGFP in S2 cells. Polo-EGFP localizes to the CS-TIPs, central spindle, and interphase MTs. Cortical contractility and membrane invagination is clearly visible in the regions where CS-TIPs contact the cortex. Frames were acquired at 15 s. intervals on a Nikon Eclipse Ti-E - TIRF microscope. The playback rate is 10 frames per second. Time: mins:secs. Scale bar, 5 µm.

DOI: https://doi.org/10.7554/eLife.38968.008

**Video 5.** *Dm* MKLP1-EGFP localization to the MT +TIPs can be obsrved throughout the cell volume. Video shows co-expression of *Dm* MKLP1-EGFP (green) and Tag-RFP-T-α-tubulin (red) in S2 cells. Spinning disk confocal microscopy was employed to visualize *Dm* MKLP1-EGFP localization towards the dorsal side of the cell. Assembly of Z-sections reveal *Dm* MKLP1-EGFP localization to the MT +TIPs towards the dorsal side and throughout the cell volume. 0.2 µm confocal z-sections were taken on a Nikon Eclipse Ti-E spinning disk confocal microscope.

DOI: https://doi.org/10.7554/eLife.38968.007

related to the fact that, unlike many other model systems, the kinase localizes robustly to interphase MTs. In cells expressing Polo-EGFP we often observed inward moving zones of Polo-EGFP exclusion (*Figure 1G*; *Video 4*). Since MTs could not penetrate these exclusion zones and were pushed inward as the zones expanded, we interpreted the Polo-EGFP exclusion zones as indicative of local regions of actomyosin contractility. Since cortical contact by Polo-positive MT plus-tips consistently triggered local contractility at both the equatorial region and in the vicinity of polar astral MT plus-ends prior to patterning, we hypothesized that: 1) Polo kinase, like centralspindlin, localizes to CS-TIPs in S2 cells, and 2) these specialized MT plus-tips trigger cortical contractility.

## Polo- and centralspindlin-positive MT plus-ends recruit cortical *Dm* ECT2 (Pebble) upon contact and activate RhoA

Since cortical contractility is triggered by the GTP-bound state of the small GTPase RhoA (*Bement et al., 2006*; *Glotzer, 2004*; *Jordan and Canman, 2012*), we next examined the behavior of the RhoA-GEF ECT2 (Pebble in *Drosophila*) by visualizing dividing cells co-expressing *Dm* ECT2-EGFP and Tag-RFP-T-α-tubulin by TIRF microscopy. During interphase *Dm* ECT2 localized to the nucleus, but a cortical pool was also evident that remained throughout mitosis. Following anaphase onset, *Dm* ECT2 localized to midzone MTs and became enriched on the nearby equatorial cortex (*Figure 2A*; *Video 6*; $n = 22/23$). *Dm* ECT2 did not robustly tip-track on astral MTs (although rare events were observed), rather, cortical *Dm* ECT2 co-localized with MT plus-tips within $10.5 \pm 1.0$ s. (mean $\pm$ SEM, $n = 21$) of contact (*Figure 2B*), which was followed by amplification of *Dm* ECT2 recruitment that peaked $1.5 \pm 0.1$ (mean $\pm$ SEM, $n = 18$) minutes after the plus-tips contacted the cortex (*Figure 2C*; *Video 6*).

We next hypothesized that MT plus-end-mediated recruitment of *Dm* ECT2 would result in local activation of RhoA. TIRF-based visualization of cells co-expressing TagRFP-T-α-tubulin and Rhotekin, a reporter for active RhoA-GTP (*Bement et al., 2005*; *Benink and Bement, 2005*), N-terminally tagged with EGFP revealed that a subset of astral MT plus-tips, which we presumed to be CS-TIPS,

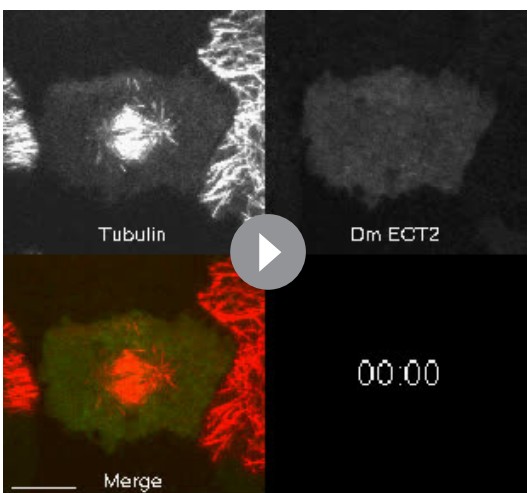

**Video 6.** Accumulation of *Dm* ECT2 (Pebble) on the CS-TIPs and its auto-amplification near the cortex. Video shows co-expression of *Dm* ECT2-EGFP (green) and Tag-RFP-T-α-tubulin (red) in *Drosophila* S2 cells. *Dm* ECT2 localization to the CS-TIPs at t = 10.15, and its amplification between t = 10.15 and 12.45 is clearly visible around the cortex. Accumulation of *Dm* ECT2-EGFP on midzone MTs and cortex is also visible. Frames were acquired at 5 s. intervals after anaphase onset on a Nikon Eclipse Ti-E - TIRF microscope. The playback rate is 20 frames per second. Time: mins:secs. Scale bar, 5 μm.

DOI: https://doi.org/10.7554/eLife.38968.010

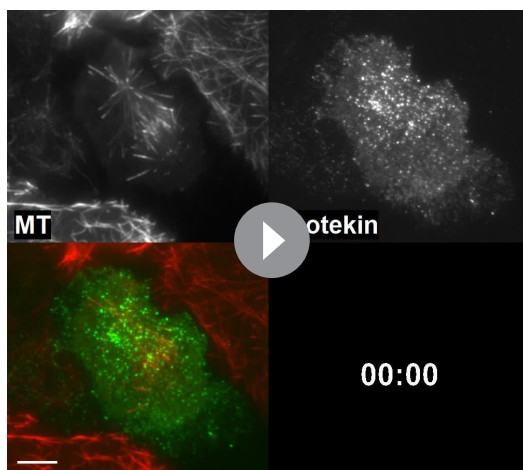

**Video 7.** CS-TIPs activate RhoA. Video shows co-expression of EGFP-Rhotekin (green) and Tag-RFP-T-α-tubulin (red) in *Drosophila* S2 cells. RhoA activation near the CS-TIPs is visible between t = 02:01 and 5:31. Cortical localization of Rhotekin-EGFP precedes its midzone localization. The midzone localization of Rhotekin-EGFP is visible between t = 5:31 and 07:31. Frames were acquired at 5 s. intervals after anaphase onset on a Nikon Eclipse Ti-E - TIRF microscope. The playback rate is 10 frames per second.Time: mins:secs. Scale bar, 5 μm.

DOI: https://doi.org/10.7554/eLife.38968.011

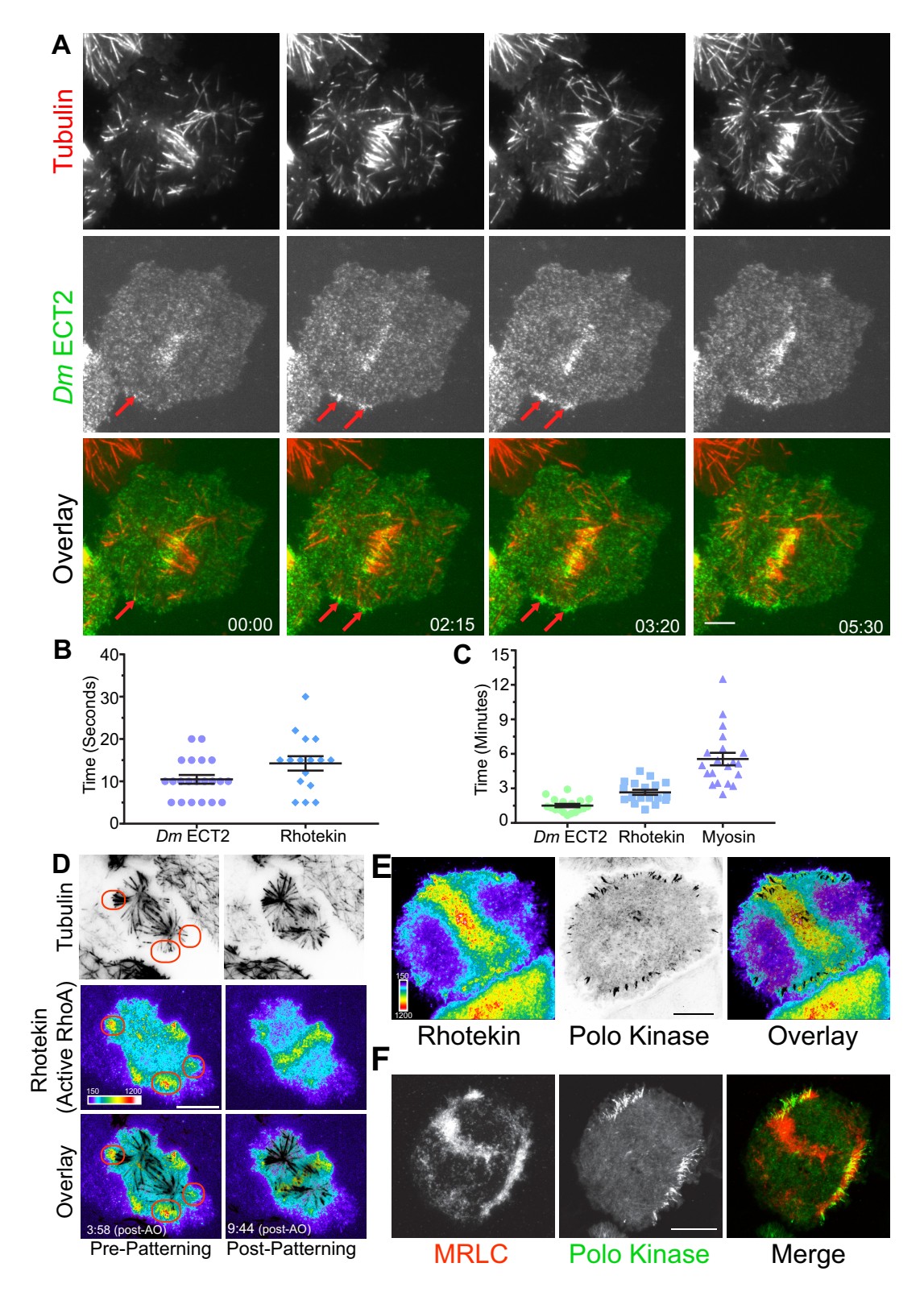

**Figure 2.** MT plus-tips recruit *Dm* ECT2 (Pebble) upon contact leading to RhoA activation and myosin accumulation. (**A**) Representative TIRF micrographs showing *Dm* ECT2 (green) recruitment to cortical locales (red arrows) physically contacted by astral MT (red) plus-tips. *Dm* ECT2 localization (time-point 00.00) is followed by auto-amplification (later time-points) near the equatorial cortex. n = 23. (**B**) Dot plots show first appearance of *Dm* ECT2 (10.48 secs ± 1.03, n = 21), and Rhotekin (14.25 secs ± 1.68, n = 16) after MTs contact the cortex. Error bars: Mean ± SEM. (**C**) Dot plots

*Figure 2 continued on next page*

*Figure 2 continued*

show average peak accumulation timings of *Dm* ECT2 (1.5 ± 0.14 mins, *n* = 18); RhoA (2.5 ± 0.2 mins, *n* = 18); and MRLC (5.5 mins ± 0.54, *n* = 19) following initial contact by astral MT plus-tips, Error Bars: Mean ± SEM. (D) Rhotekin is enriched at astral plus-tips in both polar and equatorial regions during the time period prior to CS-TIP patterning and becomes restricted to equatorial tips when patterning typically occurs. In this cell, MT plus-tip-signaling is detectable prior (3:58) to robust midzone RhoA activation (9:44). Red ovals denote polar signaling TIPs prior to patterning onto equatorial MTs. (E) Polo-positive CS-TIPs activate cortical RhoA (Rhotekin) upon contact during cytokinesis. (F) Polo-positive CS-TIPs trigger cortical contractility as evidenced by co-localized MRLC accumulation. Time: mins:secs. Color wedge displays pixel values from 150 to 1200. Scale bars, 10 µm (B, C, and D), 5 µm (A).

DOI: https://doi.org/10.7554/eLife.38968.009

locally activated RhoA within seconds of contacting the cortex (*Figure 2B,D*; *Video 7*, *n* = 18/18). The Rhotekin (active RhoA) signal initially appeared within 14.3 ± 1.7 s. of cortical contact by the astral MT plus-tips and local enrichment peaked within 2.5 ± 0.2 min. of first contact (*Figure 2B,C*; *n* = 18). Due to the high spatial and temporal resolution of the live-cell TIRF microscopy, both MT plus-tip-based and midzone-derived RhoA activation signals could be visualized in Rhotekin-expressing cells. The two activation signals often arrived simultaneously, although MT plus-tip signaling sometimes preceded evident midzone-localized RhoA activation (*Figure 2D*, *Video 7*). We confirmed that centralspindlin and Polo kinase were enriched on the MT plus-tips that triggered RhoA activation by visualizing the co-localization of cortically enriched TagRFP-T-Rhotekin with both *Dm* MKLP1-EGFP (not shown) and Polo-EGFP (*Figure 2E*; *Video 8*, *n* = 20/22).

During cytokinesis, activation of RhoA results in assembly and contraction of an actomyosin ring. We reasoned that if CS-TIPs were involved in RhoA activation near the cortex, then it would result in myosin accumulation in their vicinity. To test this hypothesis, we made a stable cell line co-expressing Polo-EGFP (to identify CS-TIPs) and myosin regulatory light chain (MRLC) tagged with TagRFP-T. Dual-color TIRF imaging of this cell line revealed a strong enrichment of MRLC near the Polo-positive MT plus-ends (*Figure 2F*; *Video 9*) that peaked within 5.5 ± 0.54 min. (*Figure 2C*; mean ± SEM, *n* = 19) of persistent CS-TIP contact with the cortex. MRLC enrichment also mediated localized cortical contractility as the CS-TIPs were clearly pulled inward when the cortical contacts persisted, which resulted in the coalescence of CS-TIPs

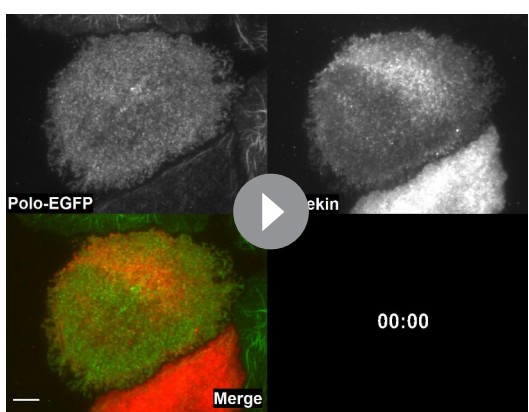

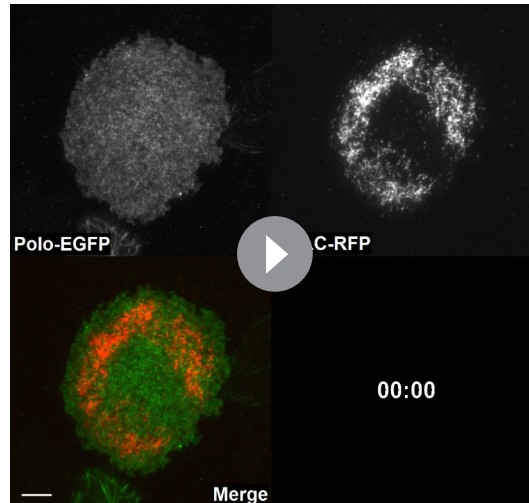

**Video 8.** Polo-positive CS-TIPs activate RhoA. Video shows co-expression of RFP-Rhotekin (red) and Polo-EGFP (green) in *Drosophila* S2 cells. In this movie, Polo-EGFP serves as a CS-TIP marker. RhoA activation near the polo positive CS-TIPs is visible between t = 01:36 and 6:36. Frames were acquired at 5 s. intervals after anaphase onset on a Nikon Eclipse Ti-E - TIRF microscope. The playback rate is 10 frames per second. Time: mins:secs. Scale bar, 5 µm.

DOI: https://doi.org/10.7554/eLife.38968.012

**Video 9.** CS-TIPs trigger cortical contractility by myosin accumulation. Video shows co-expression of myosin light regulatory chain (MRLC)-RFP (red) and Polo-EGFP (green) in *Drosophila* S2 cells. Myosin accumulation and cortical contractility near the polo positive CS-TIPs are visible between t = 02:00 and 13:16. Frames were acquired at 5 s. intervals on a Nikon Eclipse Ti-E - TIRF microscope. The playback rate is 10 frames per second. Time: mins:secs. Scale bar, 5 µm.

DOI: https://doi.org/10.7554/eLife.38968.013

within a band of enriched MRLC localized several microns interior to the initial MT contact points at the cortex (*Figure 2F*; *Video 9*). CS-TIPs were required to maintain RhoA activation (*Video 8*) and cortical contractility (*Video 9*) as polar sites of Rhotekin and MRLC enrichment were lost upon equatorial patterning and disassembly of polar Polo-positive CS-TIPs. Identical results for the establishment and maintenance of RhoA activation and MRLC enrichment were attained in cells expressing *Dm* MKLP1 (*n* = 12). Altogether, these results support the conclusion that CS-TIPs recruit *Dm* ECT2 to activate cortical RhoA thereby resulting in localized myosin accumulation and induction of cortical contractility.

## Investigating the contribution of midzone components (ABK, Polo kinase, kinesin-4, MKLP2) to CS-TIP assembly

Given the central role ABK and Polo kinase play in cytokinesis, we proceeded to investigate the contribution of ABK and Polo kinase activities to CS-TIP assembly. Wash-in experiments with the *Drosophila* specific ABK inhibitor, Binucleine 2, and the Polo kinase inhibitor BI 2536 (*Eggert et al., 2004*; *Lénárt et al., 2007*; *Smurnyy et al., 2010*) were conducted to isolate cytokinesis-specific contributions of each kinase. Polo-EGFP or *Dm* MKLP1-EGFP cells were continuously imaged from metaphase to anaphase onset and Binucleine 2 was directly introduced into the imaging chamber when CS-TIPs first appeared. Interestingly, wash-in of Binucleine 2 resulted in a complete loss of Polo kinase and *Dm* MKLP1 from MT plus-tips within 2.5 ± 0.43 min. of introducing the inhibitor (*Figure 3*, A,B; *Videos 10* and *11*; *n* = 12). Introducing the Polo kinase inhibitor BI 2536, which we and others have shown effectively inhibits *Dm* Polo kinase (*Carmena et al., 2014*; *Kachaner et al., 2017*; *Lénárt et al., 2007*), did not affect the CS-TIP localization of either Polo kinase or *Dm* MKLP1 (*Figure 3*, C,D; *Videos 12* and *13*; *n* = 10). Taken together, the data demonstrate that ABK activity is required for centralspindlin and Polo kinase to localize to CS-TIPs while Polo kinase activity is not.

PRC1 (Feo in *Drosophila*) is a well-characterized midzone MT component that preferentially bundles anti-parallel MTs (*Bieling et al., 2010*). PRC1 forms a complex with kinesin-4 (Klp3A in *Drosophila*) that regulates midzone MT length and stabilization (*Bieling et al., 2010*; *Hu et al., 2011*; *Subramanian et al., 2013*; *Zhu and Jiang, 2005*). Klp3A also localize to the central spindle and midbody. Mutations in the gene that encode Klp3A protein in *Drosophila* caused defects in central

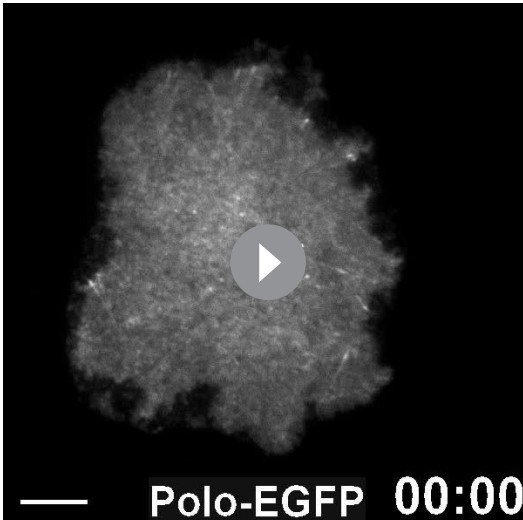

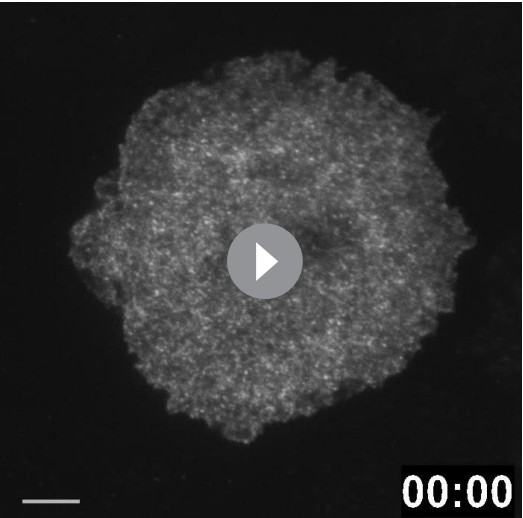

**Video 10.** ABK is required for CS-TIPs assembly. Video shows the expression of Polo-EGFP in S2 cells. Addition of ABK inhibitor, Binucleine 2, between t = 03:15 and 04:35 results in disassembly of CS-TIPs. Frames were acquired at 15 s. intervals on a Nikon Eclipse Ti-E - TIRF microscope. The playback rate is 10 frames per second. Time: mins:secs. Scale bar, 5 μm.
DOI: https://doi.org/10.7554/eLife.38968.016

**Video 11.** ABK is required for CS-TIPs assembly. Video shows the expression of *Dm* MKLP1-EGFP in S2 cells. Addition of ABK inhibitor, Binucleine 2, between t = 04:10 and 6:10 results in disassembly of CS-TIPs. Frames were acquired at 10 s. intervals on a Nikon Eclipse Ti-E - TIRF microscope. The playback rate is 10 frames per second. Time: mins:secs. Scale bar, 5 μm.
DOI: https://doi.org/10.7554/eLife.38968.017

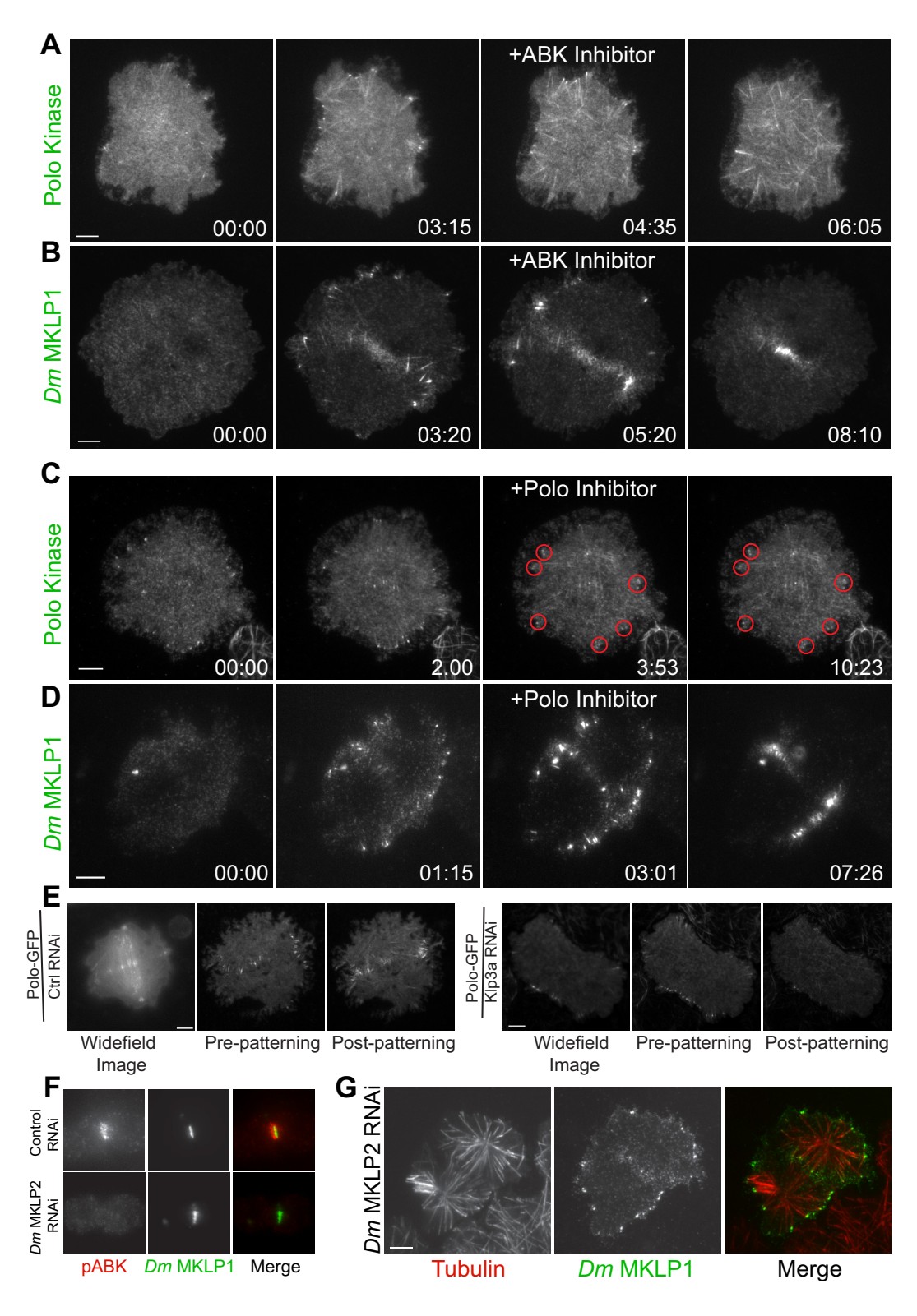

**Figure 3.** Plus-tip localization of CS-TIP components requires ABK activity, but not Polo kinase activity or midzone-localized ABK. (**A, B**) Polo and *Dm* MKLP1 are rapidly lost from CS-TIPs following wash-in of the ABK inhibitor Binucleine 2. Zero time-points indicate anaphase onset, 04:35 and 05:20 time-points indicate the time of Binucleine 2 addition. *n* = 12. (**C, D**) Polo and *Dm* MKLP1 remain localized to the CS-TIPs following wash-in of the Polo kinase inhibitor BI 2536. Zero time-points indicate anaphase onset, 03:53 and 03:01 time-points indicate the time of BI 2536 addition. *n* = 10. (**E**) Control

*Figure 3 continued on next page*

*Figure 3 continued*

cells show normal localization of Polo-EGFP on the CS-TIPs and midzone (left). Depletion of *Dm* Kinesin-4 (Klp3A) results in loss of Polo kinase from the midzone, but not from the CS-TIPs (Right). The first image is a representative frame from a wide-field movie, while last two images are representative frames from a live-cell TIRF movie for control and Klp3A depletion. *n* = 25. (**F**) Depletion of *Dm* MKLP2 leads to loss of active phosphorylated ABK (pABK) at the midzone (left panel), yet *Dm* MKLP1 assembles on CS-TIPs normally (right panel). *n* = 30. Time: mins:secs. Scale bars, 5 μm (C, D, and E), 10 μm (A, B, and F).

DOI: https://doi.org/10.7554/eLife.38968.014

The following figure supplement is available for figure 3:

**Figure supplement 1.** Klp3A depletion does not affect CS-TIPs or midzone localization of *Dm* MKLP1.

DOI: https://doi.org/10.7554/eLife.38968.015

spindle assembly during late anaphase/early telophase (*Williams et al., 1995*). Notably, defects in the central spindle assembly didn't have a major impact on spindle length and cells exhibited normal mitosis (*Williams et al., 1995*). To investigate if the PRC1-Klp3A complex contributes to CS-TIP assembly, Klp3A was depleted from Polo-EGFP or *Dm* MKLP1-EGFP expressing cells. In agreement with prior work (*D'Avino et al., 2007*), Polo kinase failed to localize to midzone MTs in cells with ≥97% of Klp3A depleted (*Figure 3E*; *Figure 3—figure supplement 1*; *Video 14*; *n* = 25); however, Polo kinase assembled normally onto CS-TIPs (*Figure 3E*). Klp3A was next depleted from cells expressing *Dm* MKLP1-EGFP and neither midzone MT localization nor CS-TIP localization was affected (*Figure 3—figure supplement 1*). These data support the conclusion that *Dm* MKLP1 and Polo kinase do not require the PRC1-Kinesin-4 complex to assemble onto CS-TIPs.

During cytokinesis, ABK localizes to midzone MTs, astral MT plus-tips, and the cortex; however, the contribution of different ABK pools to furrow formation is unclear. In many model systems, including *Drosophila*, CPC localization to the midzone is mediated by MKLP2 (Subito in *Drosophila*) (*Cesario et al., 2006*; *Gruneberg et al., 2004*; *Kitagawa et al., 2013*; *Nguyen et al., 2014*). To investigate whether midzone-localized ABK contributed to CS-TIP assembly, *Dm* MKLP2 was depleted from cells expressing *Dm* MKLP1-EGFP. Depletion of *Dm* MKLP2 resulted in a significant

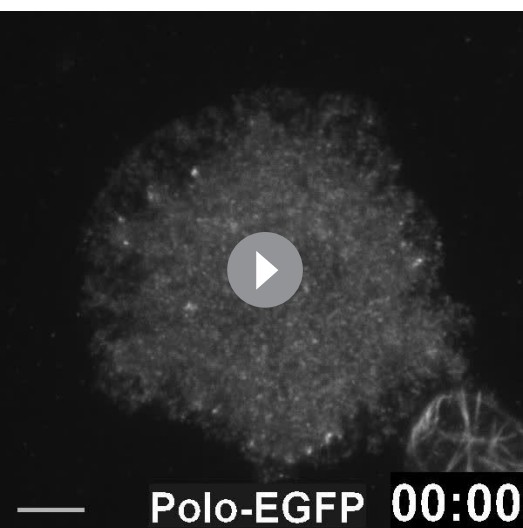

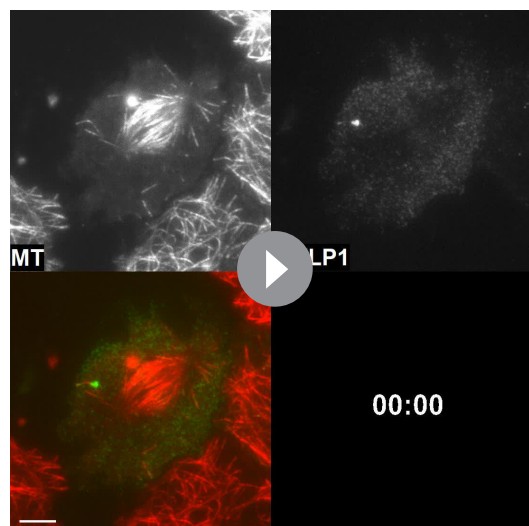

**Video 12.** Polo kinase is not required for CS-TIPs assembly. Video shows the expression of Polo-EGFP in S2 cells. Addition of Polo inhibitor, BI 2536, between t = 02:45 and 03:53, doesn't result in CS-TIPs disassembly. Frames were acquired at 15 s. intervals after anaphase onset on a Nikon Eclipse Ti-E - TIRF microscope. The playback rate is 10 frames per second. Time: mins:secs. Scale bar, 5 μm.

DOI: https://doi.org/10.7554/eLife.38968.018

**Video 13.** Polo kinase is not required for CS-TIPs assembly. Video shows co-expression of *Dm* MKLP1-EGFP (green) and Tag-RFP-T-α-tubulin (red) in S2 cells. Addition of Polo inhibitor, BI 2536, between t = 01:20 and 02:50, doesn't result in CS-TIPs disassembly. Frames were acquired at 5 s. intervals on a Nikon Eclipse Ti-E - TIRF microscope. The playback rate is 10 frames per second. Time: mins:secs. Scale bar, 5 μm.

DOI: https://doi.org/10.7554/eLife.38968.019

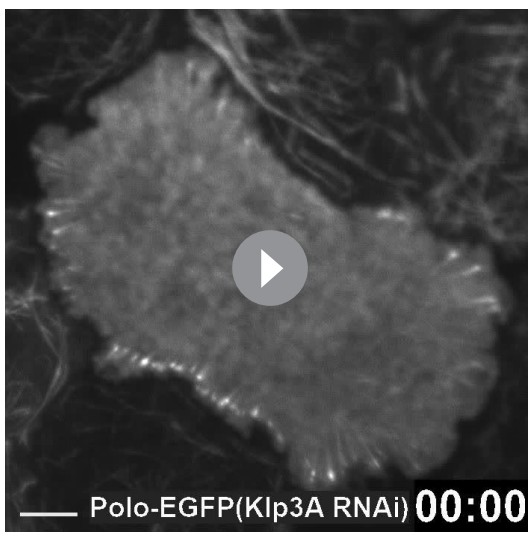

**Video 14.** Polo Kinase doesn't localize to the midzone, but assembles normally on the CS-TIPs in Klp3A depleted cells. Video shows the expression of Polo-EGFP in Klp3A depleted cells in *Drosophila* S2 cells. Polo-EGFP doesn't localize to the midzone, but assembles normally on the CS-TIPs. Frames were acquired at 5 s. intervals on a Nikon Eclipse Ti-E - TIRF microscope. The playback rate is 10 frames per second. Time: mins:secs. Scale bar, 5 μm.
DOI: https://doi.org/10.7554/eLife.38968.020

active RhoA from the vicinity of astral MT plus-tips within 3.16 ± 0.77 min. of introducing Binucleine 2 (*Video 15*; *n* = 10). Identical results were attained in cells co-expressing TagRFP-T-Rhotekin and *Dm* MKLP1-EGFP (*Video 16*; *n* = 10). Loss of localized Rhotekin and *Dm* MKLP1 from MT plus-tips occurred concomitantly and within ~2–3 min. of Binucleine 2 addition, demonstrating that ABK activity is required for both CS-TIP assembly and downstream activation of RhoA.

Polo kinase activity is required for targeting ECT2 to the midzone by promoting a physical association between ECT2 and centralspindlin (*Burkard et al., 2009*; *Petronczki et al., 2007*). Thus, we next examined the effects of Polo kinase inhibition on downstream signaling to RhoA activation by CS-TIPs. Introduction of BI 2536 to anaphase cells expressing TagRFP-T-Rhotekin and EGFP-α-Tubulin resulted in a dramatic reduction in active RhoA enrichment near the astral MT plus-tips within ~4 min. of adding the inhibitor (*Video 17*; *n* = 8). The same experiment was also conducted in cells co-expressing *Dm* MKLP1-EGFP and TagRFP-T-Rhotekin. In agreement with our prior observations (*Figure 3D*), Polo kinase inhibition did not affect

reduction in midzone levels of active phosphory-lated ABK (*Figure 3F*; *Figure 3—figure supplement 1*; *n* = 30), but *Dm* MKLP1 still localized robustly to CS-TIPs and cells completed cytokinesis normally (*Figure 3G*), (*Ye et al., 2015*). The results from the Binucleine 2 wash-ins and *Dm* MKLP2 depletions, taken together, support the conclusion that while global ABK activity is essential for CS-TIP assembly, the midzone-based ABK pool is largely dispensable. Since cortical ABK becomes evident following CS-TIP assembly (*Figure 1C*), we favor the interpretation that a soluble cytosolic pool of active ABK promotes assembly of CS-TIPs after anaphase onset.

## ABK and polo kinase activities are required for RhoA activation by CS-TIPs

We next examined the specific roles of ABK and Polo kinase in CS-TIP-mediated activation of RhoA by conducting wash-in experiments with Binucleine 2 and BI 2536. Dual color TIRF imaging of anaphase cells expressing TagRFP-T-Rhotekin and EGFP-α-tubulin revealed a dramatic loss of

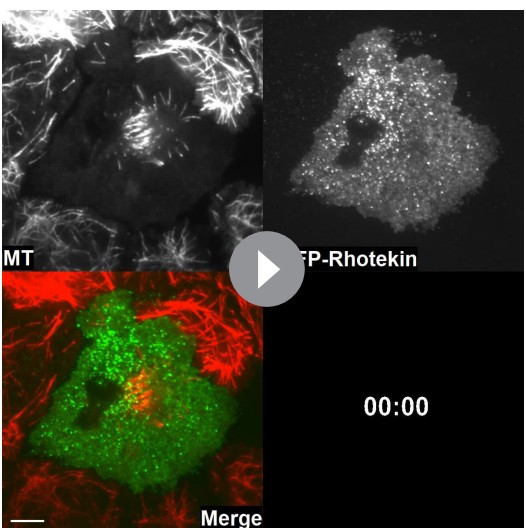

**Video 15.** ABK is required for CS-TIP signaling. Video shows co-expression of EGFP-Rhotekin (green) and Tag-RFP-T-α-tubulin (red) in *Drosophila* S2 cells. RhoA activation near the CS-TIPs is visible between t = 02:00 to 3:55. Addition of ABK inhibitor, Binucleine 2, between t = 03:55 and 5:08 results in loss of Rhotekin from the CS-TIPs within ~3 min. However, midzone Rhotekin signals take ~6.30 min. to disappear. Frames were acquired at 5 s. intervals after anaphase onset on a Nikon Eclipse Ti-E - TIRF microscope. The playback rate is 10 frames per second. Time: mins:secs. Scale bar, 5 μm.
DOI: https://doi.org/10.7554/eLife.38968.021

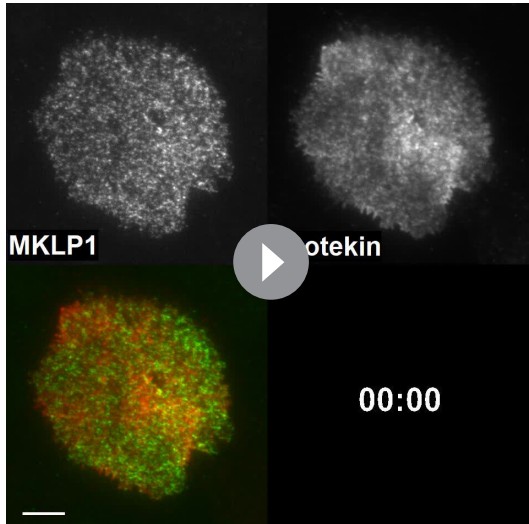

**Video 16.** ABK is required for CS-TIPs signaling. Video shows the expression of MKLP1-EGFP (green) and Rhotekin-RFP (red) in *Drosophila* S2 cells. RhoA is visible around the cortex at t = 1.00, but it gets reorganized near the CS-TIPs by t = 3:58. Addition of ABK inhibitor, Binucleine 2, between t = 04:48 and 6:30 results in loss of Rhotekin from the CS-TIPs within ~3 min. Frames were acquired at 5 s. intervals after anaphase onset on a Nikon Eclipse Ti-E - TIRF microscope. The playback rate is 10 frames per second. Time: mins:secs. Scale bar, 5 μm.
DOI: https://doi.org/10.7554/eLife.38968.022

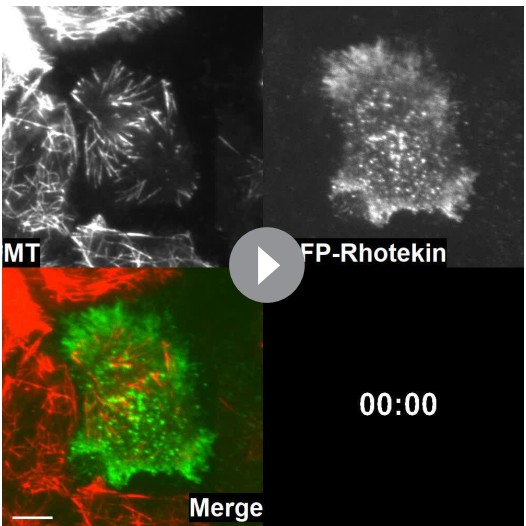

**Video 17.** Polo kinase activity is required for CS-TIPs signaling. Video shows co-expression of EGFP-Rhotekin and Tag-RFP-T-α-tubulin (red) in *Drosophila* S2 cells. RhoA activation near the CS-TIPs is visible between t = 00:00 to 01:40. Addition of Polo inhibitor, BI 2536, between t = 01:40 and 2:51 results in loss of Rhotekin from the CS-TIPs within ~3 min. Frames were acquired at 5 s. intervals on a Nikon Eclipse Ti-E - TIRF microscope. The playback rate is 10 frames per second. Time: mins:secs. Scale bar, 5 μm.
DOI: https://doi.org/10.7554/eLife.38968.023

the CS-TIP localization of *Dm* MKLP1-EGFP; however, Rhotekin signal was lost in the vicinity of the *Dm* MKLP1-positive MT plus-tips (*Video 18*; n = 8). Thus, Polo kinase activity is required for CS-TIP signaling, but not for CS-TIP assembly. We reasoned that since Polo kinase activity is required for CS-TIP-mediated activation of RhoA, then Polo inhibition should alter the localization of *Dm* ECT2 in response to physical contact by CS-TIPs. Indeed, Polo kinase inhibition resulted in the loss of localized cortical *Dm* ECT2 from CS-TIP contact points (*Video 19*; n = 4).

## Satellite MT arrays analogous to midzone and astral MTs assemble in proximity to the cortex

Over the course of visualizing hundreds of dividing cells by TIRF microscopy we frequently noticed MT structures that assembled in close proximity to the cortex as cells progressed through cytokinesis. We referred to these structures as satellite MT arrays because they formed de-novo and were not evidently linked to the spindle apparatus. Interestingly, two classes of satellite arrays were observed, which were deemed midzone and astral brush satellites because their behavior and organization were analogous to conventional midzone and astral MTs respectively.

Similar to prior observations, we noted that *Dm* MKLP1 on astral MT plus-tips bundled both parallel and anti-parallel MTs (*Vale et al., 2009*). Midzone satellites were comprised of anti-parallel bundles of equatorial astral MTs that crossed paths by growing towards the cortex, often buckling upon contact, and eventually becoming bundled by *Dm* MKLP1 and likely other cross-linking factors such as PRC1 (*Video 2*). Midzone satellites assembled in close proximity to the cortex and the ensuing cortical contractility further bundled the MTs and enriched *Dm* MKLP1 (and Polo kinase) in the overlap regions as MTs were moved inward. The process resulted in midzone satellites being in direct contact with the contractile machinery throughout cytokinesis (*Figure 4A*; *Video 2*). In most cases satellite midzones were eventually pulled into the conventional midzone that was positioned microns away from the cortex when contractility had initiated.

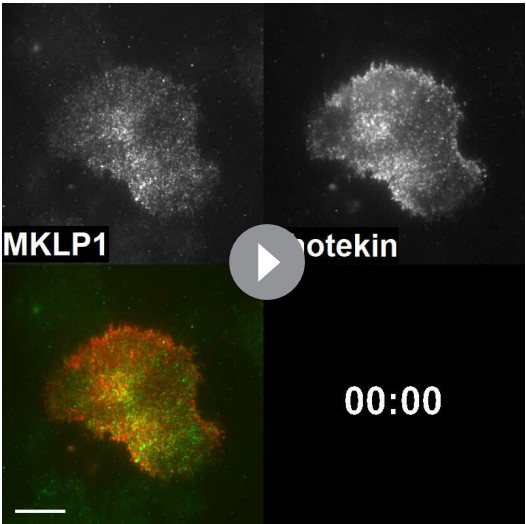

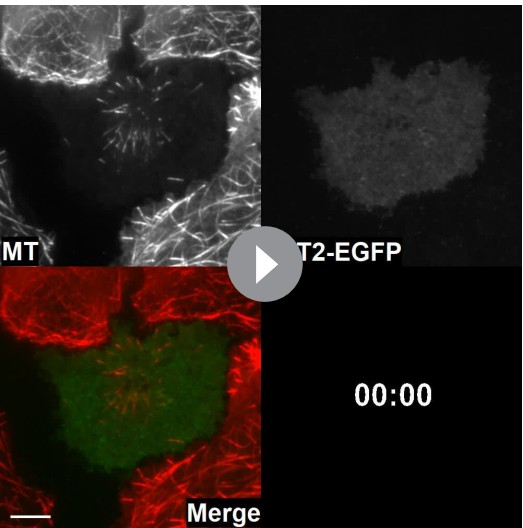

**Video 18.** Polo kinase activity is required for CS-TIPs signaling but not for assembly. Video shows the expression of MKLP1-EGFP (green) and RFP-Rhotekin (red) in *Drosophila* S2 cells. RhoA accumulation near the CS-TIPs is visible after t = 01:02. Addition of Polo inhibitor, BI 2536, between t = 01:27 and 02:55 results in loss of Rhotekin from the CS-TIPs within ~4.30 min, however MKLP1 localization to the cortex is still visible. Frames were acquired at 5 s. intervals on a Nikon Eclipse Ti-E - TIRF microscope. The playback rate is 10 frames per second. Time: mins:secs. Scale bar, 5 µm.
DOI: https://doi.org/10.7554/eLife.38968.024

**Video 19.** Polo kinase activity is required for ECT2 accumulation near the cortex. Video shows co-expression of *Dm* ECT2-EGFP (green) and Tag-RFP-T-α-tubulin (red) in *Drosophila* S2 cells. ECT2 localization on the CS-TIPs is visible after t = 04:29. Addition of Polo inhibitor, BI 2536, between t = 04:34 and 05:49 results in loss of ECT2 from the CS-TIPs within ~3 min. Frames were acquired at 5 s. intervals after anaphase onset on a Nikon Eclipse Ti-E - TIRF microscope. The playback rate is 10 frames per second. Scale bar, 5 µm.
DOI: https://doi.org/10.7554/eLife.38968.025

Astral brush satellites formed de novo from cytosolic nucleation centers producing dynamic MTs that became organized over time into branched, spindle pole-like structures with MTs oriented toward the cortex (*Figure 4B*). While exhibiting astral-like properties, astral brushes typically assembled ~5–10 microns away from the spindle apparatus and the MTs comprising astral brushes did not emanate from spindle poles or centrosome. Furthermore, the MT plus-tips of astral brushes were often enriched with centralspindlin and activated RhoA, especially prior to patterning of CS-TIPs onto equatorial MTs (*Figure 4B*).

Imaging astral brushes by TIRF microscopy was limited to the portions of MTs that were close to the plasma membrane. To better assess the properties of astral brush satellites we next employed correlative TIRF-spinning disk confocal microscopy on the same cell in order to visualize MTs outside of the TIRF field (*Figure 4C*). In this approach a cell would be imaged by time-lapse TIRF microscopy until the assembly of astral brushes became evident at which point a 2-color spinning disk confocal Z-section would be rapidly acquired. This approach revealed the existence of oblique astral brushes with MTs, only the tips of which were evident in the TIRF field, oriented at a steep angle towards the bottom of the cell (*Figure 4C* – inset 1) as well as brushes consisting of multiple MTs in the TIRF field that were oriented nearly parallel and closer to the plasma membrane (*Figure 4C* – inset 2). In analyzing both the confocal Z-slices (*Video 5*) as well as 3D reconstructions (*Videos 20* and *21*), it was evident that oblique brushes originated considerably further up into the cell volume than the astral brushes in which MTs could be seen in the TIRF field. For example, the vertex of the brush in inset 1 (*Figure 4C*) was ~4 µm from the bottom of the cell versus ~1.4 µm for the astral brush in inset 2. In both scenarios the base of the astral brush structures were >5 µm from the centrosomes and midzones thereby qualifying them as legitimate 'satellite' structures per our description of this phenomenon.

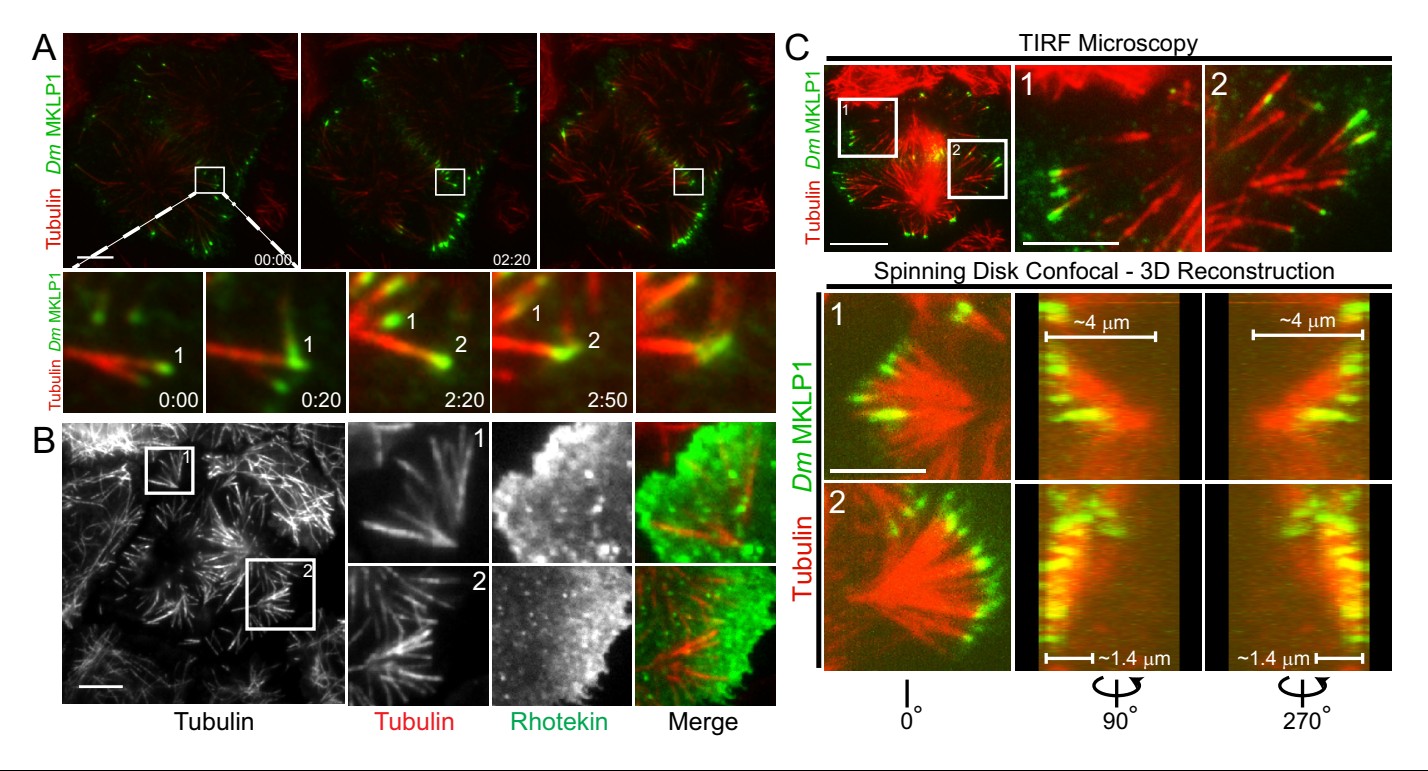

**Figure 4.** Satellite MT arrays provide additional cortical-proximal signaling platforms. (**A**) Representative frames from a TIRF time-lapse of *Dm* MKLP1-EGFP (*Video 2*), (top panel). Inset shows enlarged midzone region (bottom panel). Two CS-TIP bundling events (labeled 1 and 2) are visible at the equatorial cortex. The CS-TIPs start at oblique angles, become organized into antiparallel bundles with localized *Dm* MKLP1, and are pushed inward. (**B**) Representative frames from a TIRF time-lapse movie of Tag-RFP-T-α-tubulin and EGFP-Rhotekin (left). Inset 1 and 2 show satellite MT arrays where Rhotekin is enriched in the vicinity of MT plus-tips (right). (**C**) In the TIRF mode, MTs were visible both in and out of the TIRF field (See inset 1 and 2 of the top panel), therefore, correlative TIRF-spinning disk confocal microscopy was performed to observe the MTs that were outside of the TIRF Field (bottom panel). Lower panels display the astral brushes highlighted in inset 1 and inset 2 from different angles of the 3-D reconstruction (left is 0°, middle is 90°, and right is 270°). The 90° and 270° views revealed that the astral brush in inset 1 extends downward from a vertex ~4 µm away from the ventral surface while the brush in inset 2 originated closer (~1.4 µm) to the ventral cortex. Time: mins:secs. Scale bars, 5 µm, 10 µm.

DOI: https://doi.org/10.7554/eLife.38968.026

## EB1 is required for robust MT plus-tip localization of CS-TIP components and contributes to furrow initiation and successful completion of cytokinesis

Having observed that CS-TIPs activate RhoA via recruitment of cortical *Dm* ECT2, we next turned our attention to the mechanism by which CS-TIP components localize to plus-tips. The MT plus-end-tracking protein, EB1 is known to autonomously tip-track, and interact with other proteins to recruit them to polymerizing MT plus-ends (*Akhmanova and Steinmetz, 2008*; *Duellberg et al., 2014*; *Honnappa et al., 2009*; *Strickland et al., 2005*). EB1 was depleted by RNAi (*Figure 5—figure supplement 1*) to examine its contribution to CS-TIP assembly and signaling. Consistent with previous work (*Rogers et al., 2002*), we observed a reduction in spindle length in EB1-depleted cells compared to control cells (control RNAi: 7.0 ± 0.86 µm, EB1 RNAi: 6.1 ± 0.69 µm; $n = 73$, p=4.31 E-12); (*Figure 5—figure supplement 1*). While >80% of control cells exhibited robust CS-TIPs, EB1 depletions, which reproducibly yielded >85% knock down efficiencies (*Figure 5—figure supplement 1*), significantly compromised tip-tracking of both Polo kinase and *Dm* MKLP1 as <10% of EB1-depleted cells exhibited MT plus-tip localization of these components (*Figure 5*, A,B; *Video 22*; $n = 31$ and 41 for control and EB1 depleted cells respectively). Since EB1 depletion compromised the plus-tip localization of centralspindlin, we reasoned that it would also result in loss of RhoA activation by MT plus-tips. To test this hypothesis, we depleted EB1 in cells expressing Rhotekin-EGFP and Tag RFP-T-α-Tubulin, and performed dual color TIRF imaging. Interestingly, while midzone RhoA activation

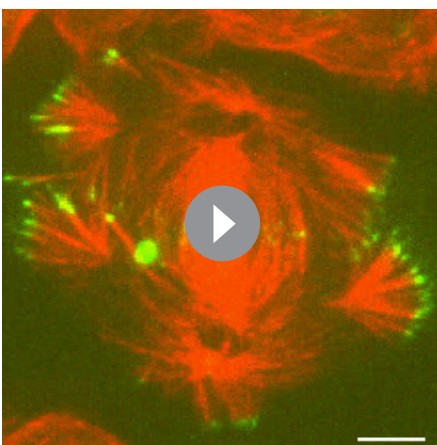

**Video 20.** Satellite MT arrays originate de novo. Video shows co-expression of *Dm* MKLP1-EGFP (green) and Tag-RFP-T-α-tubulin (red) in S2 cells. Correlative TIRF-spinning disk confocal microscopy was employed to investigate the nature and origin of satellite MT arrays. 3-D image reconstruction reveals that oblique brushes originate considerably further up into the cell volume than the astral brushes in which MTs could be seen in the TIRF field. For example, the vertex of the brush in inset 1 (see *Figure 5C*) seem to originate ~5 μm deep from the bottom of the cell. 0.2 μm confocal z-sections were taken on a Nikon Eclipse Ti-E spinning disk confocal microscope, and 3-D movie was made with the help of MetaMorph software.
DOI: https://doi.org/10.7554/eLife.38968.027

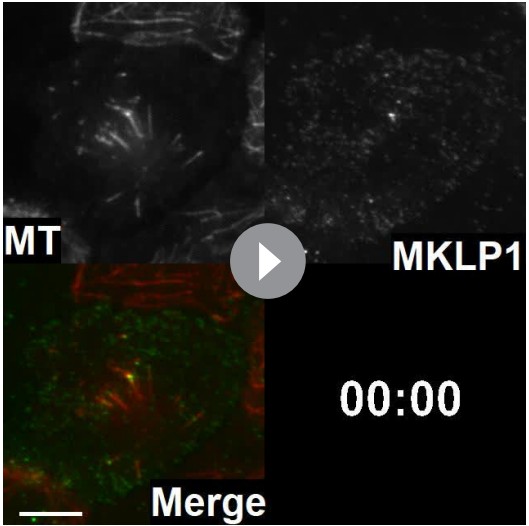

**Video 22.** EB1 is required for CS-TIPs assembly. Video shows co-expression of *Dm* MKLP1-EGFP (green) and Tag-RFP-T-α-tubulin (red) in *Drosophila* S2 cells. In contrast to control RNAi cells, depletion of EB1 in cells expressing *Dm* MKLP1-EGFP results in loss of CS-TIPs assembly. Frames were acquired at 5 s. intervals on a Nikon Eclipse Ti-E - TIRF microscope. The playback rate is 10 frames per second. Time: mins:secs. Scale bar, 5 μm.
DOI: https://doi.org/10.7554/eLife.38968.032

was detectable in nearly all EB1 depleted cells (19/21), EB1 depletion resulted in the loss of evident RhoA activity in the vicinity of MT plus-tips in 20/21 cells (*Figure 5—figure supplement 1*; *Video 23*).

Furrow initiation defects have been previously reported in dividing sea urchin eggs upon injection of either an inhibitory antibody against EB1 or a dominant negative EB1 truncation (*Strickland et al., 2005*). Furthermore, depletion of both EB1 and EB3 from HeLa cells caused a 5-fold increase in cytokinesis failure (*Ferreira et al., 2013*). To assess the functional contributions of EB1 to cytokinesis in S2 cells, we depleted EB1 from cells expressing Tag-RFP-T-α-tubulin and subjected them to overnight imaging. Interestingly, EB1 depletion resulted in furrow initiation failure in 15.5% mitotic cells, which is 4.7-fold higher than the control RNAi cells where only 3.3% mitotic cells exhibited furrow initiation failure. (*Figure 5C*; *n* = 120 and 110 for control and EB1 RNAi cells respectively). Cytokinesis failure results in binucleated cells and, accordingly, analysis of fixed cells revealed that EB1-depleted cells exhibited a ~3.5–5 fold increase in the number of binucleated cells with respect to control cells (*Figure 5D* and *Figure 5—figure supplement 1*; three independent RNAi experiments, *n* = 570 and 603 cells counted in total for control- and EB1 RNAi-treated cells respectively). Thus, EB1

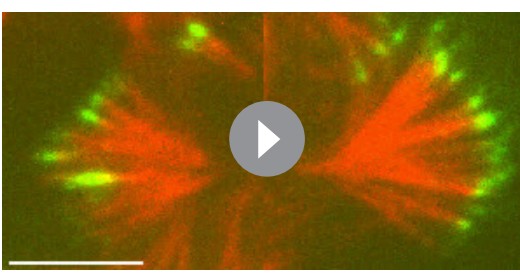

**Video 21.** 3-D view of the Satellite MT arrays. The Video gives a 3-D impression of the satellite MT-arrays of the same cell shown in *Figure 5C* (see *Video 20* and *Figure 5C*). *Dm* MKLP1-EGFP is shown in green while Tag-RFP-T-α-tubulin is shown in red. 0.2 μm confocal z-sections were taken on a Nikon Eclipse Ti-E spinning disk confocal microscope, and 3-D image reconstruction was made with the help of MetaMorph software.
DOI: https://doi.org/10.7554/eLife.38968.028

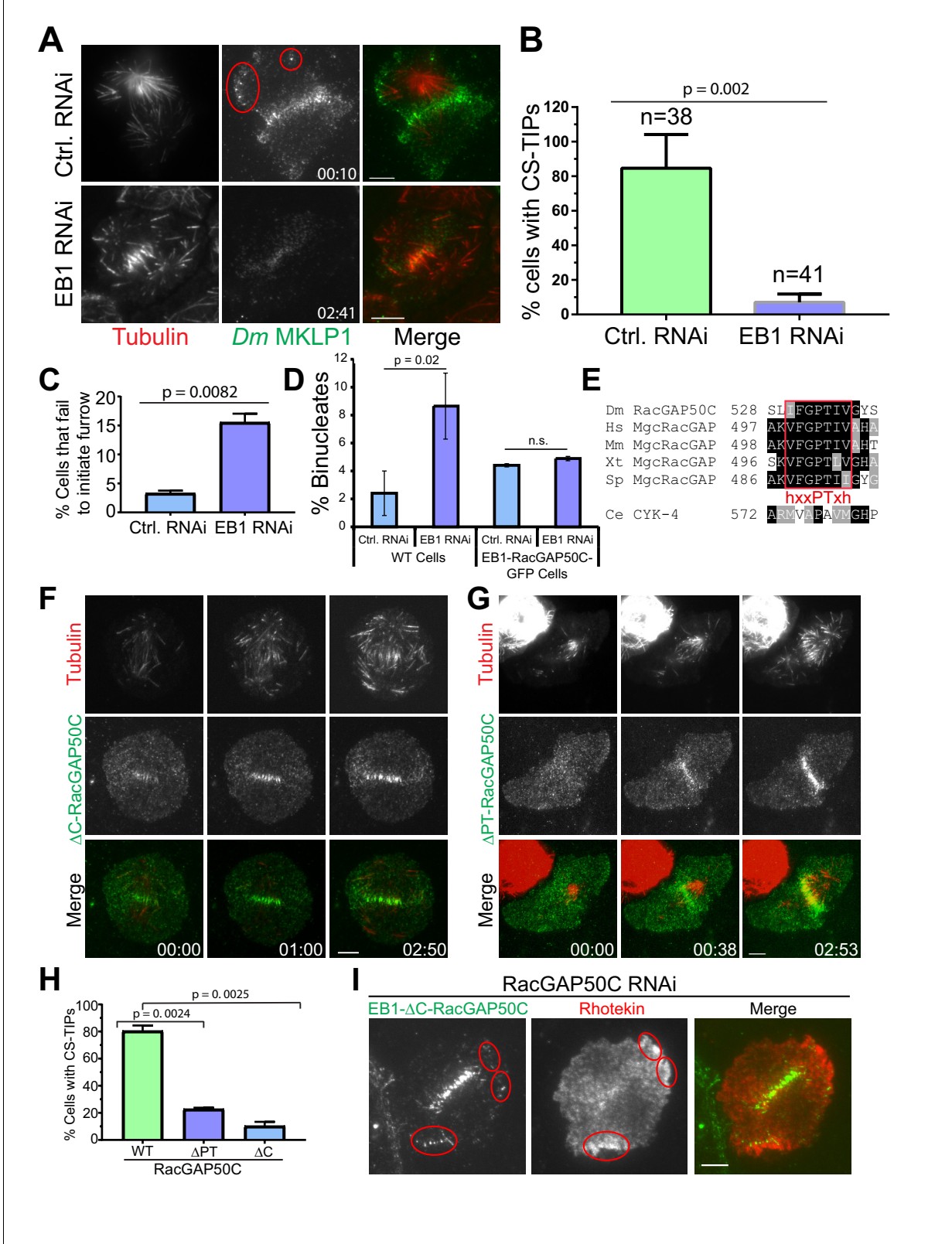

**Figure 5.** EB1 and a ''PTIV'' motif in RacGAP50C are required for robust centralspindlin localization to MT plus-tips and efficient cytokinesis. (**A**) TIRF micrographs showing loss of CS-TIPs assembly in cells depleted of EB1 (bottom panel); control cells show normal CS-TIPs localization (top panel). (**B**) Quantitation of CS-TIPs assembly in control and EB1 depleted cells (observed in both Polo and *Dm* MKLP1-EGFP-expressing cells; two experiments each, *n* = 38 and 41 for control and EB1 depleted cells respectively) (**C**) Quantitation of cells that fail to initiate furrowing in control and EB1 RNAi cells. *Figure 5 continued on next page*

Figure 5 continued
n = 120 and 110 for control and EB1 RNAi cells respectively, Error bars = Mean ± SD. (D) Quantitation of binucleated cells in control and EB1 RNAi conditions in WT cells (n = 3 independent experiments; n = 570 and 603 cells counted in total for control- and EB1 RNAi-treated cells respectively) and cells expressing an EB1-RacGAP50C-EGFP hybrid (n = 2 independent RNAi experiments; n = 431 and 550 cells counted in total for control- and EB1 RNAi-treated cells respectively). (E) The putative hxxPTxh motif (IFGPTIV) in RacGAP50C is highly conserved although key residues in the motif are different in *C. elegans*. (F, G) Deletion of C-terminus or ''PTIV motif'' in RacGAP50C disrupts MT plus-tip enrichment, but not midzone or cortical localization. (H) Quantitation of CS-TIPs assembly in cells expressing WT (n = 21), ΔPTIV (n = 26), or ΔC-RacGAP50C-EGFP (n = 20), three experiments each, Error bars: Mean ± SD. (I) Still frames from live-cell TIRF microscopy showing RhoA activation in the vicinity of CS-TIPs in a cell expressing EB1-Δ C-RacGAP-EGFP fusion protein and depleted of endogenous RacGAP50C. Student's t-test two-tailed p-values are reported, not significant (n.s.) is a p-value>0.05. Error bars: Mean ± SD. Scale Bars, 5 μm.
DOI: https://doi.org/10.7554/eLife.38968.029

The following figure supplements are available for figure 5:

**Figure supplement 1.** Effects of EB1 depletion on spindle length, RhoA activation, and cytokinesis.
DOI: https://doi.org/10.7554/eLife.38968.030
**Figure supplement 2.** Investigating contribution of centralspindlin to spatial RhoA activation cues.
DOI: https://doi.org/10.7554/eLife.38968.031

depletion compromised CS-TIP assembly and signaling, and resulted in functional consequences as cells did not initiate furrow ingression and failed cytokinesis at higher rates than control cells.

We next tested if restoring centralspindlin to MT plus-tips could rescue the effects of EB1 depletion by generating cells expressing an EB1-RacGAP50C-EGFP hybrid. EB1-RacGAP50C-EGFP constitutively tip-tracked on polymerizing MTs and this behavior was retained when endogenous EB1 was depleted by RNAi targeting the 3'UTR of EB1, which was not present in the hybrid sequence (*Figure 5—figure supplement 1*). While depletion of EB1 resulted in an increase in binucleated WT cells, there was not a statistically significant difference in the percentage of binucleates between control and EB1 depleted cells expressing EB1-RacGAP50C-EGFP (*Figure 5D* and *Figure 5—figure supplement 1*; two independent RNAi experiments, n = 431 and 550 cells counted in total for control and 3' UTR EB1 RNAi cells respectively).

## MT plus-tip localization of RacGAP50C requires its C-terminus and a new type of EB1-interaction motif

Our characterization of the EB1-dependent nature of CS-TIP tracking is consistent with recent work in *Xenopus laevis* embryonic cells describing the plus-tip tracking behavior of the centralspindlin component MgcRacGAP via a SxIP EB1-interaction motif during cytokinesis (*Breznau et al., 2017*). The SxIP motif in *X. laevis* MgcRacGAP is not conserved beyond *Xenopus* species despite the fact that centralspindlin plus-tip tracking is clearly conserved in *Drosophila* and human cells. Our group recently identified unconventional EB1-interaction motifs in the *Drosophila* chromokinesin NOD (deemed 'PT' motifs) (*Ye et al., 2018*) that are similar to a newly characterized MT plus-tip localization sequence (*Kumar et al., 2017*; *Manatschal et al., 2016*; *Stangier et al., 2018*). From our work and others, we propose a PT motif consensus of hxxPTxh ('h', hydrophobic a.a.; x, any amino acid). Sequence alignments revealed a conserved, with the possible exception of *C. elegans*, putative hxxPTxh motif in the

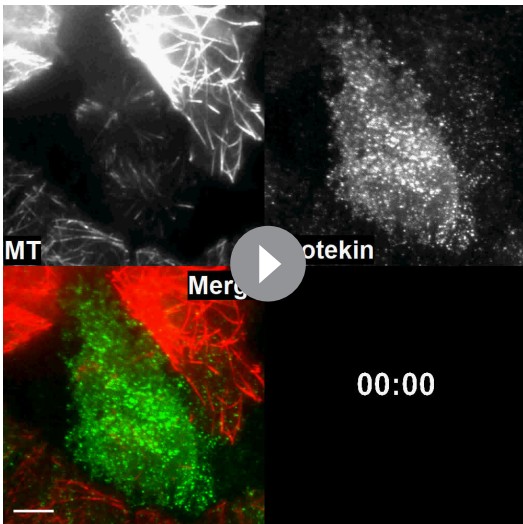

**Video 23.** Depletion of EB1 results in loss of RhoA activation and focusing near the CS-TIPs. Video shows co-expression of EGFP-Rhotekin (green) and Tag-RFP-T-α-tubulin (red) in *Drosophila* S2 cells. EB1 depletion results in loss of active RhoA near the CS-TIPs but not at the midzone. Frames were acquired at 5 s. intervals on a Nikon Eclipse Ti-E - TIRF microscope. The playback rate is 10 frames per second. Time: mins:secs. Scale bar, 5 μm.
DOI: https://doi.org/10.7554/eLife.38968.033

C-terminal region of MgcRacGAP/Cyk-4 (*Figure 5E*). After observing RacGAP50C localization on CS-TIPs (*Figure 1B*), we generated a C-terminal truncation of RacGAP50C (1-526) lacking the hxxPTxh motif (a.a. 530–536; IFGPTIV), but retaining the ability to associate with *Dm* MKLP1 via its N-terminus (*Mishima et al., 2002*; *Pavicic-Kaltenbrunner et al., 2007*). ΔC-RacGAP50C resulted in a severe reduction in tip-tracking efficiency; however, its localization to the midzone and equatorial cortex was unaffected (*Figure 5F,H*; *Video 24*; *n* = 20). We next generated a more targeted deletion mutant of RacGAP50C in which only four amino acids ($PTIV^{533-536}$) were deleted from the hxxPTxh motif. The PTIV deletion does not remove any amino acids that are required for GAP activity and structural simulations (*Biasini et al., 2014*) did not reveal a significant change in the structure of the deletion mutant (*Figure 5—figure supplement 2*). Like the ΔC-RacGAP50C mutant, ΔPTIV-RacGAP50C localized normally to midzone MTs and the equatorial cortex (*Figure 5G*; *Video 25*; *n* = 26). However, the tip-tracking activity of ΔPTIV-RacGAP50C was compromised as the mutant exhibited tip-tracking in only ~23% of cells that were transfected compared to ~80% of cells expressing WT RacGAP50C (*Figure 5H*). It should be noted that the RacGAP50C mutants had to be studied through transient transfections because attempts to make stable cell lines expressing the ΔC or ΔPTIV RacGAP50C mutants were unsuccessful.

## RacGAP50C mutants fused to EB1 support MT plus-end based activation of RhoA

Since deletion of either the C-terminus or hxxPTxh motif in RacGAP50C compromised the MT plus-tip localization of RacGAP50C, we sought to further assess centralspindlin functionality when RacGAP50C localization was restored to plus-tips via direct fusion to EB1. Unlike prior failed attempts to make stable cell lines expressing the ΔC–RacGAP50C mutants, we were able to make stable cell lines expressing the mutants fused to EB1. Depletion of *Dm* MKLP1, which results in co-depletion of RacGAP50C (*D'Avino et al., 2006*), resulted in the absence of obvious cortical regions of localized RhoA activation in 11/13 *Dm* MKLP1-depleted cells due to an apparent loss of both midzone derived

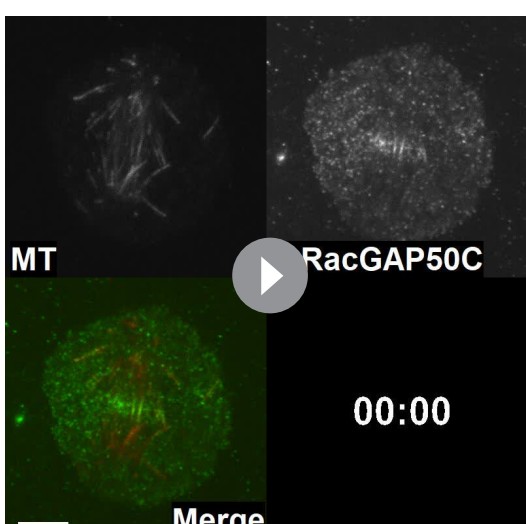

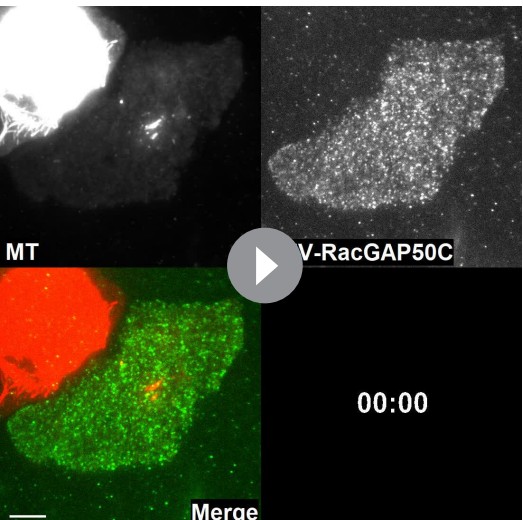

**Video 24.** C-terminal truncation of RacGAP50C results in loss of CS-TIPs assembly. Video shows co-expression of ΔC-RacGAP50C-EGFP (green) and Tag-RFP-T-α-tubulin (red) in *Drosophila* S2 cells. In contrast to full length RacGAP50C, C-terminal truncation of RacGAP50C results in loss of CS-TIPs assembly. Frames were acquired at 5 s. intervals after anaphase onset on a Nikon Eclipse Ti-E - TIRF microscope. The playback rate is 10 frames per second. Time: mins:secs. Scale bar, 5 μm.
DOI: https://doi.org/10.7554/eLife.38968.034

**Video 25.** Deletion of ''PTIV'' motif in RacGAP50C results in loss of CS-TIPs assembly. Video shows co-expression of ΔPTIV-RacGAP50C-EGFP (green) and Tag-RFP-T-α-tubulin (red) in *Drosophila* S2 cells. In contrast to full length RacGAP50C, deleion of ''PTIV'' motif in RacGAP50C results in loss CS-TIPs assembly. Frames were acquired at 5 s. intervals after anaphase onset on a Nikon Eclipse Ti-E - TIRF microscope. The playback rate is 10 frames per second. Time: mins:secs. Scale bar, 5 μm.
DOI: https://doi.org/10.7554/eLife.38968.035

and MT plus-tip-based spatial cues (*Figure 5—figure supplement 2*; *Video 26*). Interestingly, depletion of EB1 resulted in the specific loss of MT plus-tip-proximal, but not midzone-based, RhoA activation (*Figure 5—figure supplement 1*). When the ΔC-RacGAP50C mutant was fused to EB1, it constitutively localized to growing plus-tips throughout the cell cycle and also localized to the midzone during anaphase/cytokinesis. Interestingly, the EB1-ΔC-RacGAP50C hybrid supported MT plus-tip-based RhoA activation in the absence of endogenous RacGAP50C in 3/3 cells that were imaged (*Figure 5I*, *Figure 5—figure supplement 2*). The finding indicates that the RhoA activating capacity of CS-TIP-based centralspindlin, likely via *Dm* ECT2 recruitment, is mediated by the N-terminal 526 a.a. of RacGAP50C.

## Discussion

In this study we report that key cytokinesis regulators localize to and track polar and astral MT plus-ends within 2–3 min. of anaphase onset (*Figure 6A,B*); however, within ~10 min. the regulators were lost from most of the polar astral MTs and maintained on equatorial MTs via a mechanism that remains to be characterized (*Figure 6C*). Rapid (seconds time-scale) recruitment and subsequent amplification of cortical *Dm* ECT2 (Rho GEF) was observed at cortical sites contacted by Polo kinase- and centralspindlin-positive MT plus-ends, which then resulted in localized activation of RhoA within seconds, and accumulation of myosin-regulatory light chain (MRLC) if the contacts persisted for minutes. Global ABK activity was required for CS-TIP assembly and signaling, while Polo kinase activity was required for signaling but not assembly. Robust plus-tip localization of Polo kinase and centralspindlin as well as RhoA activation in the vicinity of MT plus-tips required EB1. The C-terminus of RacGAP50C and a newly identified EB1-interaction motif contained therein contributed to centralspindlin plus-tip tracking suggesting that the interaction between EB1 and centralspindlin may be mediated by a conserved hxxPTxh motif and potentially other motifs in the C-terminus of RacGAP50C. We propose that CS-TIPs act as signaling hubs that aid in cleavage furrow positioning by physically delivering a plus-tip associated population of centralspindlin that rapidly recruits cortical *Dm* ECT2 contact sites - triggering RhoA activation and localized cortical contractility (*Figure 6D*).

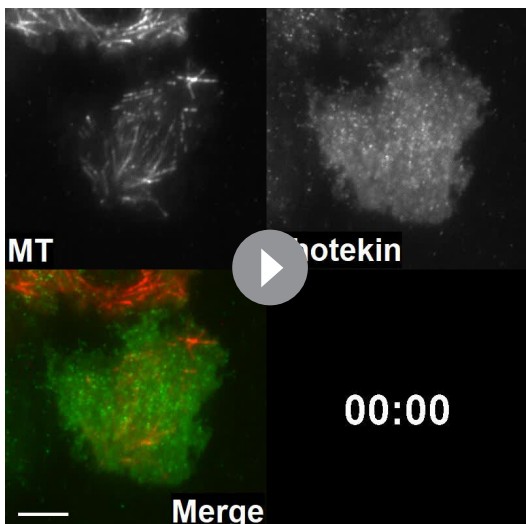

**Video 26.** Depletion of *Dm* MKLP1 results in loss of RhoA activation and focusing near the CS-TIPs and midzone. Video shows co-expression of EGFP-Rhotekin (green) and Tag-RFP-T-α-tubulin (red) in *Drosophila* S2 cells. In contrast to WT cells, *Dm* MKLP1 depletion results in loss of active RhoA near the CS-TIPs and midzone. Frames were acquired at 5 s. intervals on a Nikon Eclipse Ti-E - TIRF microscope. The playback rate is 10 frames per second. Time: mins:secs. Scale bar, 5 µm. Time: mins:secs. Scale bar, 5 µm.
DOI: https://doi.org/10.7554/eLife.38968.036

### CS-TIP- vs midzone-based signaling in positioning the cleavage furrow

The positioning of the contractile region has been variously proposed to involve positive equatorial cues derived from the central spindle, astral MTs, or both; and inhibitory factors acting at poles (*Canman et al., 2003*; *Mogilner and Manhart, 2016*). Direct evidence of RhoA activation by astral MTs is lacking. Here we show that *Dm* ECT2 is recruited to cortical CS-TIP contact sites within seconds, which, in turn, leads to rapid localized RhoA activation in the vicinity of MT plus-tips (*Figure 2*). How do the molecular constituents of a CS-TIP, namely Polo kinase and centralspindlin, carry out this function (*Figure 6D*)? It is presently unknown which component of a CS-TIP is responsible for physically recruiting cortical *Dm* ECT2, but based on prior work (*Burkard et al., 2009*; *Petronczki et al., 2007*; *Somers and Saint, 2003*; *Wolfe et al., 2009*), we postulate that it is mediated by the N-terminus of RacGAP50C once it has been phosphorylated by Polo kinase. Our data also show that the C-terminus of RacGAP50C and EB1

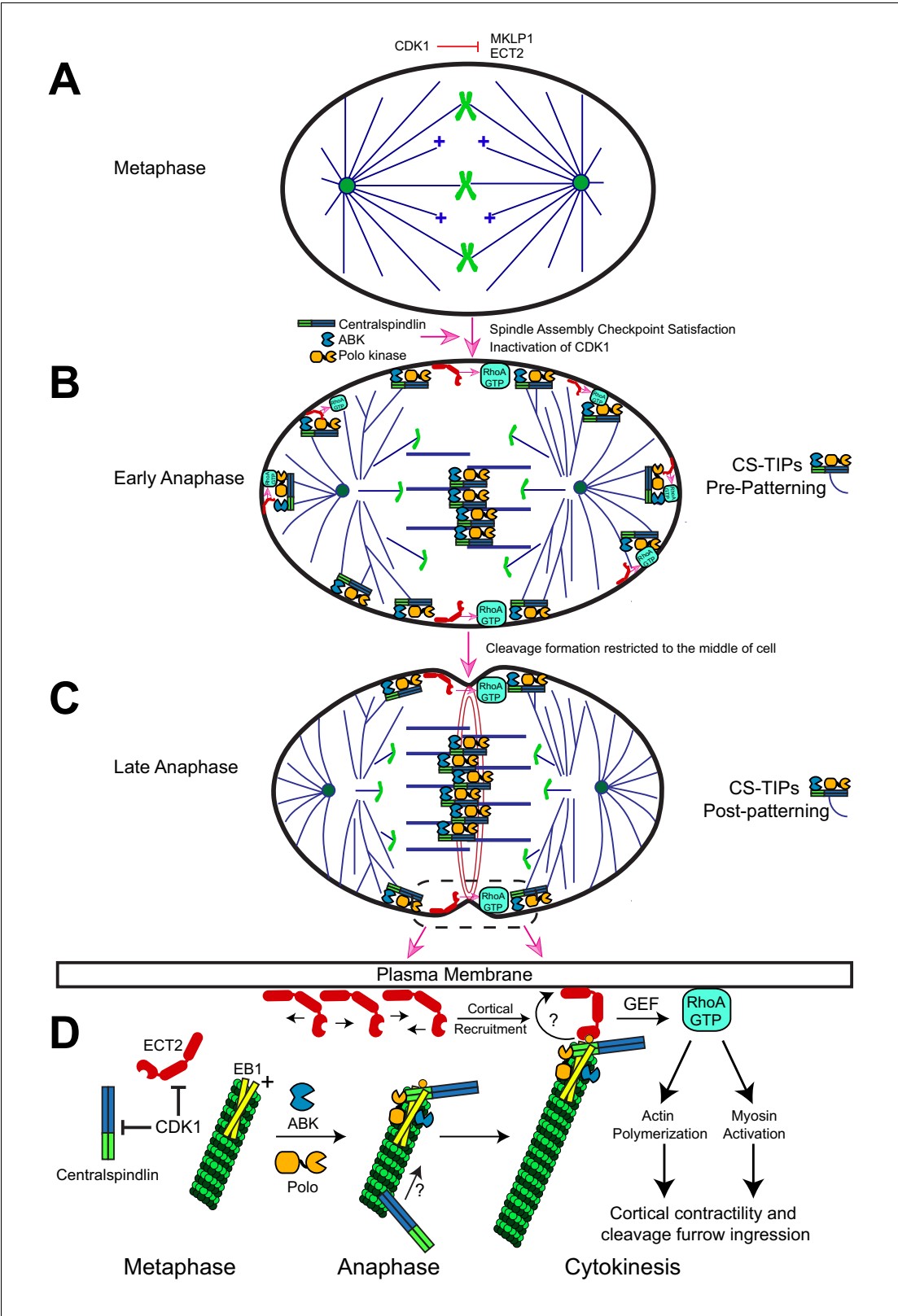

**Figure 6.** Description of the CS-TIP patterning phenomenon and a model for how CS-TIPs activate RhoA during cytokinesis. (**A**) Phosphorylation of MKLP1 by CDK1 during metaphase impedes its association with the spindle MTs. (**B**) As cells progress into anaphase, CDK1 activity drops, resulting in loss of MKLP1 phosphorylation and binding of MKLP1 to spindle MTs. Within 2–3 min. of anaphase onset, the centralspindlin complex, ABK, and Polo kinase decorate the plus-tips of polar and equatorial astral MTs. CS-TIPs locally activate RhoA by recruiting ECT2 after physically contacting the cortex.

*Figure 6 continued on next page*

*Figure 6 continued*

(C) A phenomenon that we refer to as CS-TIP patterning occurs ~10 min. post anaphase onset and involves loss of CS-TIPs from the polar astral MTs and retention on the equatorial MT plus-tips thereby supporting equatorial assembly of an actomyosin ring. (D) Model for how CS-TIPs trigger cortical contractility. High CDK1 activity in metaphase prevents centralspindlin binding to MTs and ECT2 enrichment on the plasma membrane. When CDK1 activity drops ECT2 becomes enriched at the plasma membrane and centralspindlin, ABK, and Polo kinase assembles onto CS-TIPs. Robust MT plus-tip localization of Polo kinase and centralspindlin requires EB1. The plus-tip-localization of centralspindlin also requires the C-terminus of RacGAP50C and a putative EB1 interaction motif therein. The plus-end directed motility of *Dm* MKLP1 may also contribute to centralspindlin's tip localization (arrow and question mark). Polo-phosphorylated (yellow circle) RacGAP50C on CS-TIPs directly binds to cortical ECT2 leading to its rapid recruitment to contact sites and amplification via unknown feedback mechanisms (curved arrow and question mark). The GEF activity of ECT2 locally generates active RhoA-GTP at the cortex leading to actin polymerization and myosin activation to promote cortical contractility and cleavage furrow ingression.

DOI: https://doi.org/10.7554/eLife.38968.037

are required for centralspindlin's plus-tip enrichment. What then is the necessity of the other central-spindlin component *Dm* MKLP1 to CS-TIP based signaling? An appealing model (discussed further below) is that the motor domain of *Dm* MKLP1 is also necessary, although not sufficient, for robust plus-end localization of centralspindlin. As for the role of plus-tip localization of Polo kinase, it is unclear whether its plus-tip localization is necessary for CS-TIP-based signaling or if the kinase's plus-end localization during cytokinesis is even conserved beyond *Drosophila*. Although, global Polo kinase activity is required for CS-TIP-based signaling. It is conceivable that centralspindlin is the only constituent of CS-TIPs necessary to recruit cortical ECT2 and that this role is conserved in cell types where centralspindlin has been observed at MT plus-ends during cytokinesis. Once ECT2 is recruited to a CS-TIP contact point, more of the GEF becomes locally amplified via an unknown process that may involve a poorly understood positive feedback mechanism (*Bement et al., 2015*; *Graessl et al., 2017*; *Tyson et al., 2003*).

Due to the high spatial and temporal resolution of the TIRF-based imaging assays we employed, both MT plus-tip-proximal and midzone-derived RhoA activation signals could be observed in Rhote-kin-expressing cells. The CS-TIPs and midzone-derived RhoA activation signals typically 'arrived' simultaneously, although CS-TIP-based signaling sometimes preceded evident midzone-derived RhoA activation (*Figure 2D*). The observation of distinct RhoA activating signals is in accordance with the notion that the MT plus-tip-based pathway described here functions concurrently with a fur-row positioning signal(s) that can 'act at a distance' to activate RhoA and myosin such as has been described in cells in which astral MTs have been depleted (*von Dassow et al., 2009*). Reliance on a diffusional midzone-derived signal to the cortex could be problematic for the fidelity of cytokinesis as the nature of such a signal would be inexorably linked to variable features of the midzone MT array such as distance from the cortex as well as the proper localization and levels of gradient-pro-ducing midzone components. In contrast, we propose that CS-TIP-based signaling should provide more consistent spatial cues than midzone-mediated signals since it would depend primarily on MT polymerization rates, lengths, and the cortical orientation of astral MTs (emanating from centro-somes, spindle poles, or satellite astral brushes) – features that are highly reproducible between bipolar anaphase spindles. Nonetheless, given the essential nature of cytokinesis, most cells likely employ both plus-tip-based (action via direct physical contact) and midzone-derived (action at a dis-tance) RhoA activation pathways.

Interestingly, in drug-triggered monopolar cytokinesis, numerous cytokinesis regulators (PRC1, MKLP1, ECT2, Plk1) localized to MT tips (*Hu et al., 2008*). The work provided further evidence that spindle bipolarity is not required for assembly of a MT array capable of positioning the cleavage fur-row even in the absence of conventional midzones with antiparallel overlap MTs. We observed a comparable localization pattern of key cytokinesis factors on astral MT plus-tips in the unperturbed bipolar cytokinesis events studied here. Thus, our proposed mechanism of RhoA activation by regu-latory factors on MT plus-tips is likely a key contributor to furrow positioning in both experimentally-induced monopolar cytokinesis and naturally occurring bipolar cytokinesis.

A midzone derived ABK gradient has been proposed to contribute to proper spatial regulation of RhoA activation (*Fuller et al., 2008*; *Tan and Kapoor, 2011*); and in addition to ABK, Polo kinase has also been shown to induce furrow initiation by recruiting ECT2 to the central spindle (*Petronczki et al., 2007*), indicating that both ABK and Polo kinase activities are required for furrow positioning. We show that CS-TIP assembly and signaling was unaffected by treatments that mis-

localize midzone pools of active ABK (*Figure 3*, F and G) and Polo kinase (*Figure 3E*). While global inhibition of either ABK or Polo kinase leads to severe cytokinesis defects, furrow positioning and cytokinesis was not evidently impacted in this study by mis-localization of midzone pools of either kinase (*Figure 3*). Thus, midzone-based ABK and Polo kinase activity gradients are either dispensable for furrow positioning in *Drosophila* or redundant with MT plus-end–based furrow positioning cues and/or other pathways.

Our conclusion that MT plus-tips activate RhoA by targeting a cortical pool of ECT2 are generally consistent with a recent study in which artificial targeting of ECT2 to the plasma membrane rescued cytokinesis defects of ECT2 depletion. Although this study also found that furrow positioning and ingression were unaffected in cells expressing an ECT2 mutant that failed to localize to midzones as a result of reduced binding to MgcRacGAP (*Kotýnková et al., 2016*). On its face, this finding contradicts our proposal that recruitment of cortical ECT2 by CS-TIPs requires centralspindlin - specifically RacGAP50C (*Figure 6D*); however, the ECT2 mutant developed in this study exhibited reduced, but still detectable enrichment at the equatorial cortex and, importantly, MgcRacGAP was still required for successful cytokinesis in cells expressing the ECT2 mutant. In light of our model, it is possible that a cortical population of the ECT2 mutant used in this study may still bind to MgcRac-GAP with sufficient affinity to support furrowing upon MT contact (*Basant and Glotzer, 2018*).

## Establishment and maintenance of cleavage furrow positioning signals

Successful cytokinesis requires that the cleavage furrow not only be established in the correct position, but also maintained over tens of minutes. During our high-resolution imaging, we often observed ''satellite MT arrays'' that were in close proximity to the cortex and assembled after contractility had been established. Near the equatorial cortex, midzone satellites were built from equatorial astral MTs that became bundled and forced into anti-parallel MT arrays by the contractile machinery. Importantly, once contractility was established, the midzone satellites maintained direct physical contact with the cortex such that any midzone-based cues would no longer have to act over long distances to signal to the cortex. The other type of satellite MT array - astral brushes – also assembled after contractility had begun, but via apparent branching MT nucleation from astral MTs. The astral brushes, in which MT plus-ends were oriented toward the cortex, locally activated RhoA by creating a high density of new CS-TIPs. We posit that midzone and astral brush MT satellites could fulfill two critical roles during cytokinesis as autonomous signaling centers that can 1) act at distances far from their analogous structures (midzones and poles/centrosomes) in the spindle proper, and 2) maintain contractility after it has been established. It is worth noting that in large cells amplification of an astral-like MT array (*Helmke et al., 2013*; *Ishihara et al., 2016*; *Mitchison et al., 2012*), likely similar to the satellite astral brushes observed here (*Figure 4, A-C*), would be required for MT tips to act over distances of tens to hundreds of microns.

## Role of EB1 and a conserved hxxPTxh motif in RacGAP50C in cytokinesis

We observed that depletion of the tip-tracking protein EB1 resulted in furrow ingression defects and increased incidence of cytokinesis failure. Our results are generally consistent with earlier reports where injection of an inhibitory antibody against EB1 or a dominant negative EB1 truncation into dividing sea urchin eggs delayed furrow ingression (*Strickland et al., 2005*), and the finding that EB1 and EB3 co-depletion in HeLa cells resulted in a 5-fold increase in cytokinesis failure (*Ferreira et al., 2013*). However, differences in the extent of mitotic phenotypes in more recent knockout studies of EB proteins in HeLa cells have been reported. (*McKinley and Cheeseman, 2017*; *Yang et al., 2017*). McKinley et al. reported spindle abnormalities in ~35% of mitotic cells upon double knockout of EB1 and EB3; however, Yang et al. reported seemingly normal mitotic spindles in cells where all three EB proteins (EB1, EB2, and EB3) were knocked out. Since these studies didn't focus intently on cytokinesis it would be worthwhile to further investigate the contribution of EB1 to cytokinesis keeping in mind that redundant pathways likely exist and that the relative contribution of these pathways may differ between model systems and cell types. Experimental conditions that isolate the MT plus-end-based pathway from redundant pathways in EB depleted/knockout cells would be informative.

*Xenopus laevis* MgcRacGAP localized to MT plus-ends in embryonic cells during cytokinesis in a C-terminal SxIP motif-dependent manner (*Breznau et al., 2017*). Interestingly, the SxIP motif identified in *X. laevis* MgcRacGAP was not conserved beyond *Xenopus* species despite the fact that centralspindlin plus-tip localization occurs in flies and human cells. Thus, there are likely other EB1 interfaces in the centralspindlin complex. In this study, we report that deletion of the C-terminus of RacGAP50C compromised MT plus-tip localization and we identified a conserved motif (hxxPTxh) in RacGAP50C that also reduced the plus-end localization of RacGAP50C when deleted. The hxxPTxh motif in RacGAP50C is similar to recently identified EB1-interaction motifs in budding yeast Kar9, fission yeast Dis1/TOG, and the *Drosophila* chromokinesin NOD (*Kumar et al., 2017*; *Manatschal et al., 2016*; *Matsuo et al., 2016*; *Ye et al., 2018*). The characteristics of centralspindlin and NOD have notable similarities and differences. Both NOD and centralspindlin localize to polymerizing MT plus-ends and contain motor domains, albeit it is noteworthy that NOD exhibited evident MT plus-end directed motility along MTs (*Ye et al., 2018*) while motile centralspindlin puncta were only occasionally observed in this study. The region in NOD containing the non-conventional EB1-interaction motifs and the NOD motor domain were each necessary but not sufficient for tip-tracking in cells. Similarly, it is possible that both the *Dm* MKLP1 motor domain and EB1 interaction motifs in RacGAP50C are required to support robust centralspindlin plus-tip-localization (*Figure 6D*). Finally, the hxxPTxh motifs in NOD are in a region of the protein that is predicted to be disordered, which is ideal for supporting higher affinity interactions with EB1. To the contrary, the hxxPTxh motif in RacGAP50C is contained within a structured region of the protein's GAP domain. While our data show that this motif contributes to MT plus-tip localization we cannot conclude that it directly interacts with EB1 nor do we rule out that other EB1 interaction motifs exist in the C-terminus of RacGAP50C or in other CS-TIP components. Thus, a more thorough investigation of EB1-CS-TIP component interactions is warranted.

## CS-TIP assembly, signaling, and patterning onto equatorial MTs

The observation that *Dm* MKLP1, RacGAP50C, ABK, and Polo kinase localize to the plus-ends of polar and equatorial astral MT tips before becoming patterned onto equatorial astral MTs raises an important question: how do CS-TIPs selectively disassemble from the polar astral MTs? Since we have shown that CS-TIPs activate cortical contractility, any patterning mechanism that disassembles polar CS-TIPs would inherently contribute to the polar relaxation phenomenon. A recent study in *C. elegans* found that TPXL-1 (*C. elegans* TPX-2 homologue)-based activation of Aurora A kinase contributed to the clearance of anillin and F-actin (*Mangal et al., 2018*) from the polar cortex. A polar Aurora A kinase activity gradient that regulates kinetochore-MT attachments exists in *Drosophila* and mammalian cells (*Chmátal et al., 2015*; *Ye et al., 2015*) although its role in cytokinesis is unclear. In *Drosophila*, a kinetochore based PP1 phosphatase activity gradient derived from a PP1-SDS22 complex has been proposed to regulate polar relaxation (*Kunda et al., 2012*; *Rodrigues et al., 2015*). It is entirely possible that phosphatase and/or kinase activity gradients in *Drosophila* contribute to the loss of CS-TIPs from polar astral MTs. Alternatively, a chromosome-based RanGTP gradient has been shown to locally reduce Anillin at the cortex (*Kiyomitsu and Cheeseman, 2013*), and could also impede CS-TIPs-based signaling from the polar astral MTs as the chromosomes approach the polar cortex in late anaphase, but prior to nuclear envelope reformation. While we presently favor the hypothesis that polar- or chromosome/kinetochore-derived activity gradients regulate CS-TIP patterning, a detailed investigation is necessary to resolve the question of how polar CS-TIPs are selectively disassembled as cells progress through cytokinesis.

## Materials and methods

**Key resources table**

| Reagent type (Species) Or resource | Designation | Source or reference | Identifiers | Additional information |
|---|---|---|---|---|
| Cell line (*Drosophila melanogaster*) | *Drosophila Schneider* 2 (S2) cell line | American Type Culture Collection | RRID:CVCL_Z232 Cat# CRL-1963 | |

*Continued on next page*

*Continued*

| Reagent type (Species) Or resource | Designation | Source or reference | Identifiers | Additional information |
|---|---|---|---|---|
| Transfected construct (*Drosophila melanogaster*) | *Dm* MKLP1-EGFP and Tag-RFP-T-α-tubulin | This study | RRID:CVCL_UZ77 | All constructs were transfected in *Drosophila Schneider* 2 (S2) cells. |
| Transfected construct (*Drosophila melanogaster*) | RacGAP50C-EGFP and Tag-RFP-T-α-tubulin | This study | RRID:CVCL_UZ78 | |
| Transfected construct (*Drosophila melanogaster*) | Polo Kinase-EGFP | This study | RRID:CVCL_UZ79 | |
| Transfected construct (*Drosophila melanogaster*) | EGFP-Rhotekin and Tag-RFP-T-α-tubulin | This study | RRID:CVCL_UZ80 | |
| Transfected construct (*Drosophila melanogaster*) | RFP-Rhotekin and *Dm* MKLP1-EGFP | This study | RRID:CVCL_UZ81 | |
| Transfected construct (*Drosophila melanogaster*) | RFP-Rhotekin and MRLC-EGFP | This study | RRID:CVCL_UZ82 | |
| Transfected construct (*Drosophila melanogaster*) | *Dm* ECT2-EGFP and Tag-RFP-T-α-tubulin | This study | RRID:CVCL_UZ83 | |
| Transfected construct (*Drosophila melanogaster*) | EB1-RacGAP50C-EGFP and Tag-RFP-T-α-tubulin | This study | RRID:CVCL_UZ84 | |
| Transfected construct (*Drosophila melanogaster*) | EB1-ΔC-RacGAP50C-EGFP and Tag-RFP-T-α-tubulin | This study | RRID:CVCL_UZ85 | |
| Transfected construct (*Drosophila melanogaster*) | EB1-ΔPTIV-RacGAP50C-EGFP and Tag-RFP-T-α-tubulin | This study | RRID:CVCL_UZ86 | |
| Antibody | Anti-Pavarotti (rabbit serum) | Kind gift from Dr. David Glover, University of Cambridge | PMID: 9585508 | 1:5000 dilution was used for western blots |
| Antibody | Anti-RacGAP50C (rabbit polyclonal) | Kind gift from Dr. David Glover, University of Cambridge | PMID: 17032738 | 1:4000 dilution was used for western blots |
| Antibody | Anti-alpha Tubulin (DM1 A) (mouse monoclonal) | Sigma-Aldrich | Cat# T6199 | 1:10,000 dilution was used for western blots |
| Antibody | Anti-EB1A (rabbit polyclonal) | Kind gift from Dr. Stephen Rogers, UNC Chapel Hill | | 1:5000 dilution was used for western blots |
| Antibody | Anti-Klp3A (rabbit polyclonal) | Kind gift from Dr. Jonathan Scholey, UC Davis | | 1:500 dilution was used for western blots |
| Chemical compound, drug | Binucleine 2 | Sigma | Cat# B1186 | |
| Chemical compound, drug | BI 2536 | Millipore Sigma | Cat# 533936 | |
| Software, algorithm | MetaMorph | MetaMorph | | |
| Software, algorithm | ImajeJ,Fiji | NIH | | |
| Software, algorithm | MS office | Microsoft | | |
| Software, algorithm | Prism | GraphPad | | |

## Cloning and site directed mutagenesis

The *Drosophila* MKLP1 gene (known as Pavarotti in *Drosophila*, CG1258) and ECT2 (known as Pebble in *Drosophila*, CG8114) were PCR amplified from cDNA clones RE22456 and SD01796 respectively with a 5′ SpeI site and a 3′ XbaI site. The resulting PCR products of the individual genes were then inserted into the 5′ SpeI and 3′ XbaI sites of the pMT/V5 His-B vector (Invitrogen) containing in-frame EGFP gene at the 3′ end, cloned between 5′ XbaI and 3′ SacII sites, and the CENPC promoter at the 5′ end, cloned between single Kpn1 site. The regulatory light chain of the non-muscle type two myosin (Spaghetti squash in *Drosophila*, CG3595) was amplified from the cDNA clone LD14743. The resulting PCR product was cloned between 5′ SpeI and 3′ XbaI of the pMT vector containing in-frame RFP gene at the 3′ end and Mis12 promoter at the 5′ end. EGFP-Rhotekin was PCR amplified from the pCS2-eGFP-rGBD plasmid (gift from the Bement lab (UW-Madison) via Wadsworth lab

(UMass Amherst)). The resulting PCR product was cloned between 5' XhoI and 3'ApaI sites of the pMT vector containing CENPC promoter at the 5' end cloned between single KpnI site. A second version of Rhotekin was also created in the pMT vector where we swapped the EGFP fluorophore with RFP. Polo kinase under its own promoter was cloned between 5' KpnI and 3' SpeI sites of the pMT vector containing in-frame EGFP gene, cloned between 5' SpeI and 3' ApaI sites. The Rac-GAP50C (also known as Tumbleweed in *Drosophila*, CG13345) gene was amplified from genomic DNA of *Drosophila*. The resulting PCR product was cloned in a similar fashion as described for MKLP1. A four-amino acid deletion (*PTIV*[533-536]) in the RacGAP50C gene was generated by Q5 site directed mutagenesis kit (NEB, MA, USA), following manufacturer's protocol. C-terminal truncation of the RACGAP50C gene was created by amplifying amino acids (1-526) from the full length Rac-GAP50C (a.a. 1–625) gene. The resulting PCR product was inserted into the 5' SpeI and 3' XbaI sites of the pMT vector containing in-frame EGFP gene at the 3' end, and the CENPC promoter at the 5' end. EB1-RacGAP50C-GFP fusion was created by inserting EB1 PCR product (amplified from EB1 cDNA) into the SpeI site of the existing RacGAP50C-GFP construct. EB1 fusions to the RacGAP50C mutants (ΔC and ΔPT) were created in a similar way. See *Table 1* for the primers used in the cloning.

## *Drosophila* Schneider (S2) cell culture and generation of transient/stable cell line

*Drosophila* Schneider (S2) cell line was obtained from American Type Culture Collection (ATCC). Identity of the cell line was confirmed by genomic DNA isolation and PCR amplification of many genes, and cells were periodically screened for mycoplasma contamination. All cell lines were grown at 25°C in Schneider's medium (Life Technologies, Carlsbad, CA), containing 10% heat-inactivated fetal bovine serum (FBS) and 0.5X antibiotic-antimycotic cocktail (Gibco, Life Technologies, NY, USA). Cell lines were generated by transfecting the DNA constructs containing the gene of interest with Effectene transfection reagent (Qiagen, Hilden, Germany), following the manufacturer's protocol. Four days after transfection, expression of EGFP/RFP tagged proteins was checked by fluorescence microscopy. To make a stable cell line, cells were selected in the presence of Blasticidin S HCl (Thermo Fisher Scientific, Waltham, MA) and/or Hygromycin (Sigma-Aldrich) until there was no observable cell death. Thereafter, cell lines were either frozen down or maintained in the S2 media @ 25°C without Blasticidin and/or Hygromycin B. Transient cell lines were imaged after 4–5 days of transfection, and they were not subjected to antibiotics selection. ABK-GFP and mCherry-α-tubulin cell line, was a generous gift from Dr. Eric Griffis, University of Dundee.

**Table 1.** Primers for cloning (5' to 3')

| | |
|---|---|
| MKLP1FW | TTCAATTGCAAATGGTACCTACTAGTATGAAGGCAGTACCCAGGAC |
| MKLP1 RV | TCCTCGCCCTTGCTCACCATTCTAGAGATTTTCGACTTCTTGCTGC |
| RacGAP50C FW | GTTTCAATTGCAAATGGTACCTACTAGTATGGCGCTCTCCGCATTGGC |
| RacGAP50C RV | CTCCTCGCCCTTGCTCACCATTCTAGATTTCTTGTGCGCAGATGCCG |
| ΔC-RacGAP50C RV | CTCCTCGCCCTTGCTCACCATTCTAGAGTTATCGATCGGCATTAGCAC |
| ΔPTIV-RacGAP50C FW | GGCTACTCCACACCAGATCC |
| ΔPTIV-RacGAP50C RV | GCCAAAGATGAGCGAGATGTTATCG |
| Polo Kinase FW | GGGGTACCGAAAAGCTGTATCGAGTCGC |
| Polo Kinase RV | GGACTAGTTGTGAACATCTTCTCCAGCA |
| Tag RFP Rhotekin FW | GGCATGGACGAGCTGTACAAGACTAGTTCCGGACTCAGATCTCGAG |
| Tag RFP Rhotekin RV | GGGATAGGCTTACCTTCGAACCGCGGTTAGCCTGTCTTCTCCAGCACCTG |
| EGFP Rhotekin FW | TTCAAGTTTCAATTGCAAATGGTACCACCATGGTGAGCAAGGGCGA |
| EGFP Rhotekin RV | TAGGCTTACCTTCGAACCGCGGGCCCGCCTGTCTTCTCCAGCACCT |
| MRLC FW | TTCAATTGCAAATGGTACCTACTAGTATGTCATCCCGTAAGACCGC |
| MRLC RV | AGCTCTTCGCCCTTAGACACTCTAGACTGCTCATCCTTGTCCTTGG |

DOI: https://doi.org/10.7554/eLife.38968.038

**Table 2.** Primers for RNAi (5' to 3')

| Subito FW | TAATACGACTCACTATAGGGCTGAAGCTAATCAATGGCAGC |
|---|---|
| Subito RV | TAATACGACTCACTATAGGGTTTTCTGAACTGTACTGGCCG |
| EB1 FW | GAATTAATACGACTCACTATAGGGAGAATGGCTGTAAACGTCTACTCCACAAATGTG |
| EB1 RV | GAATTAATACGACTCACTATAGGGAGATGCCCGTGCTGTTGGCACAGGCGTTTA |
| EB1 3'UTR FW | TAATACGACTCACTATAGGGTTTGAAACGTGAACGAAAACCCAC |
| EB1 3'UTR RV | TAATACGACTCACTATAGGGAAACGACAAAAGTCAGCTAGTGAAA |

DOI: https://doi.org/10.7554/eLife.38968.039

## RNA interference (RNAi) experiments

Around 500 base pairs of DNA templates containing T7 promoter sequence at the 5' end for *Dm* MKLP2 (also known as Subito in *Drosophila*, CG12298), EB1 (CG3265), and *Dm* Kinesin-4 (also known as Klp3A in Drosophila, CG8590) were generated by PCR. Double-stranded RNAs (dsRNAs) were synthesized from the respective DNA templates at 37°C using the T7 RiboMax Express Large-Scale RNA Production System (Promega Corp., Madison, WI), following the manufacturer's protocol. For RNAi experiments, cells at about 25% confluency were incubated in a 35 × 10 mm tissue culture dish for an hour. Thereafter, media was carefully aspirated off the dish and 1 ml of serum-free S2 media containing 20 $\mu$g of dsRNA was added to the dish. After 1 hr, 1 ml of fresh S2 media containing FBS was added to the dish and incubated for 2–4 days at 25°C. Depletion of endogenous Rac-GAP50C was achieved by PCR amplification of 270 base pairs of 3' UTR and 200 base pairs of the coding sequence. Depletion of the endogenous EB1 was carried out by amplification of 600 bp of the EB1 3'UTR from a cDNA clone (FI12926 – DGRC). Following PCR amplification, double-stranded RNAs (dsRNAs) were synthesized as described above, and 20 $\mu$g of dsRNA was used for RNAi experiments. Cells were analyzed after 4 days of incubation by quantitative western blotting and binucleates counting. See *Table 2* for primer sequences.

## Immunofluorescence

*Drosophila* S2 cells were plated on an acid-washed Concanavalin A (Sigma-Aldrich) coated coverslips for 30 min, then cells were quickly rinsed with 1X BRB80 buffer. Subsequently, cells were fixed with 10% paraformaldehyde for 10 min. Cells were then permeabilized with phosphate-buffered saline (PBS) containing 1% Triton X-100 for 8 min, rinsed three times with PBS plus 0.1% Triton X-100, and blocked with 5% boiled donkey serum ((Jackson Immuno Research Laboratories, Inc, West Grove, PA) for 60 min. All primary antibodies were diluted in 5% boiled donkey serum. Anti-Phospho-aurora A/B/C (Cell Signaling Technology, Danvers, MA) was used at a concentration of 1:1000, and anti-tubulin antibody (DM1A; Sigma-Aldrich) at 1:2000. All secondary antibodies (Jackson Immuno Research Laboratories, Inc, West Grove, PA) were diluted in boiled donkey serum at 1:200. After secondary antibody incubation, coverslips were washed three times with PBS plus 0.1% Triton X-100, followed by incubation with 4',6'-diamidino-2-phenylindole (DAPI) at a concentration of 1:1000 for 10 min, and three additional washes with PBS plus 0.1% Triton X-100. Coverslips were sealed in a mounting media containing 20 mM Tris, pH 8.0, 0.5% N-propyl gallate, and 90% glycerol. Four-color Z-series consisting of ~30 planes at 0.2 $\mu$m intervals were acquired for GFP, RFP, Cy5, and DAPI channels on a TIRF-Spinning Disk Confocal system assembled on a Nikon Ti-E microscope equipped with a 100 × 1.4 NA DIC Apo Oil immersion objective, two Hamamatsu ORCA-Flash 4.0 LT digital CMOS camera (C11440), four laser lines (447nm, 488nm, 561nm, and 641nm), and a Meta-Morph software. Fluorescence intensities were obtained by drawing a 50 × 100 grid manually around the maximum-intensity projection of the Z-series images. The local background was estimated by placing the 50 × 100 region to a nearby place. To obtain the ratio intensities, identical regions were drawn manually on the *Dm* MKLP1-EGFP channel and transferred to a phosphorylated ABK (pABK) channel.

## Quantitative western blotting

Equal amount of proteins was loaded into a 10% or 8% SDS-PAGE gel. After running the gel, proteins were transferred to a nitrocellulose membrane using the Trans-Blot Turbo transfer system (Bio-

Rad Laboratories, Inc, Hercules, CA) for 10 min. Subsequently, the membranes were incubated in 5% milk (V/V made in Tris-buffered saline with 0.1% Tween) for one hour. Following blocking, the membranes were incubated with their respective primary antibodies (α-EB1 - 1:5000, α-Klp3A - 1:500, α-Tubulin – 1:10,000, α-Pavarotti/*Dm* MKLP1 – 1:5,000, α-RacGAP50C – 1:4,000) for 1 hr, followed by three times wash and secondary antibody incubation at 1:5000 dilution for 1 hr. Following secondary antibody incubation, membranes were washed three times in Tris-buffered saline with 0.1% Tween (TBST) and imaged with a G:BOX system controlled by GeneSnap software (Syngene, Cambridge, U.K.). Images were further quantified to estimate the knock down efficiency with the help of Fiji/Image J software (Schindelin et al., 2012). To obtain the intensity values, identical regions were drawn over all the bands and their integrated intensity was recorded. Intensity values were normalized to their respective loading controls (Tubulin) to estimate the knockdown efficiency.

## Fluorescence and live-cell total internal reflection fluorescence (TIRF) microscopy

Around 500 to 700 μl of cells, expressing the gene of interest were seeded on a 35 mm glass bottom dish with 20 mm bottom well (Cellvis, CA, USA) coated with Concanavalin A for 30–60 min. Before imaging, 2 ml of fresh S2 media containing FBS was added to the dish. Live-cell TIRF movies were acquired on a Nikon Ti-E microscope equipped with a 100 × 1.4 NA DIC Apo Oil immersion objective, a Hamamatsu ORCA-Flash 4.0 LT digital CMOS camera (C11440), four laser lines (447nm, 488nm, 561nm, and 641nm), and a MetaMorph software. Live-cell TIRF movies of all the constructs (MKLP1-EGFP and Tag-RFP-T-α-tubulin; RacGAP50C-EGFP and Tag-RFP-T-α-tubulin; Polo Kinase-EGFP; ABK-GFP and Tubulin mCherry; EGFP-Rhotekin and Tag-RFP-T-α-tubulin; Tag-RFP-T-Rhotekin and *Dm* MKLP1-EGFP; Tag-RFP-T-Rhotekin and MRLC-EGFP; *Dm* ECT2-EGFP and Tag-RFP-T-α-tubulin; EB1-RacGAP50C-EGFP and Tag-RFP-T-α-tubulin; EB1-ΔC-RacGAP50C-EGFP and Tag-RFP-T-α-tubulin; EB1-ΔC-RacGAP50C-EGFP and Tag-RFP-T-Rhotekin; EB1-ΔPTIV-RacGAP50C-EGFP and Tag-RFP-T-α-tubulin) were acquired at the following settings: 561 exposure time 200 ms, 488 exposure time 200 ms, temperature 25°C, and frame acquisition rate 5 s. To observe cytokinesis defects, overnight imaging was performed on control and EB1 depleted cells. Cells were seeded on Concanavalin A coated 35 mm glass bottom dish as described above. A total of 76 time-points were acquired for 50 stage positions at the interval of 4 min @ 25°C on a Nikon Eclipse Ti-E microscope (Nikon, Tokyo, Japan) equipped with 40X/1.30 Oil Nikon Plan Fluor DIC N2 objective (Nikon), Andor iXon$_3$ EMCCD camera (Andor Technology, Belfast, U.K.), and Metamorph software (Molecular Devices, Sunnyvale, CA). Images were further analyzed with the help of MetaMorph software.

## Spinning disk confocal and correlative TIRF-spinning disk confocal imaging

Cells were seeded on Concanavalin A (Con A) coated glass-bottom dishes. After 30–45 min. of incubation, cells were imaged on a TIRF-Spinning Disk system assembled on an Eclipse Ti-E inverted microscope (Nikon), equipped with a Borealis (Andor Technology, Ltd., Belfast, U.K.) retrofitted CSU-10 (Yokogawa Electric Corp., Tokyo, Japan) spinning disk head and two ORCA-Flash4.0 LT Digital CMOS (Hamamatsu Corp., Bridgewater, NJ), using a 100 × 1.49 NA Apo differential interference contrast objective (Nikon). Metamorph software was used to control the imaging system. For correlative TIRF-spinning disk confocal imaging, cells were followed by time-lapse TIRF imaging; once the cell had assembled CS-TIPs (based on localization of *Dm* MKLP1-GFP on MT +TIPs) the system was switched to spinning disk mode and 0.2 μm confocal z-sections were taken. All quantifications and 3D-reconstructions were made with the help of Metamorph software.

### Chemical perturbation experiments

Cells expressing *Dm* MKLP1-EGFP and Tag-RFP-T-α-Tubulin; Polo-EGFP; Rhotekin-EGFP and Tag-RFP-T-α-Tubulin; Rhotekin-RFP and *Dm* MKLP1-EGFP; *Dm* ECT2-EGFP and Tag-RFP-T-α-Tubulin were subjected to ABK or Polo inhibition. Cells were seeded on Concanavalin A coated 35 mm glass bottom dish with 20 mm bottom well (Cellvis, CA, USA) for 30–60 min. Before imaging, 2 ml of fresh S2 media containing FBS was added to the dish. Live-cell TIRF movies were acquired on a Nikon Ti-E microscope as described above. The whole field of view under the microscope was scanned to find mitotic cells. Mitotic cells expressing the gene of interest were imaged at the interval of 20–30 s.

until the anaphase onset. After anaphase onset, when the CS-TIPs were visible, S2 media in the glass bottom dish was exchanged with S2 media containing Binucleine 2 (40µM) for ABK inhibition or BI 2536 (1µM) for Polo inhibition. Thereafter, imaging was continued at 5 s. intervals. Images were further analyzed with the help of Metamorph software (CA, USA).

### GFP/RFP intensity quantification

GFP/RFP intensities before and after the addition of inhibitor (Binucleine two or BI 2536) were quantified to estimate the effects of inhibitor on CS-TIPs patterning and signaling. Identical regions were drawn over the GFP/RFP puncta to measure the fluorescence intensity. The local background was estimated by placing the same region to a nearby place. Background intensity values were subtracted from GFP/RFP values to estimate the actual GFP/RFP fluorescence intensity. Histograms and dot plots were generated using Prism software (Graphpad, CA, USA), and figures were assembled using Adobe Illustrator software (Adobe, CA, USA). Statistical analysis was performed using Prism software (Graphpad, CA, USA).

## Acknowledgments

Thank you to Margaux Audett, UMass, for help with generating reagents. We acknowledge Patricia Wadsworth, UMass and Alex Mogilner, NYU for sharing insightful conversations. We are also tremendously grateful to Eric Griffis, University of Dundee, for the kind gift of ABK cell line; Bill Bement, UW-Madison for the kind gift of Rhotekin plasmid via Patricia Wadsworth; Stephen Rogers, UNC Chapel Hill for the kind gift of α-EB1A antibody; Jonathan Scholey, UC Davis for the kind gift of α-Klp3A antibody; and David Glover, University of Cambridge for his generosity in sharing α-Pavarotti and RacGAP50C antibodies. This work was supported by an NIH grant (GM107026) to TJM.

## Additional information

### Funding

| Funder | Grant reference number | Author |
|---|---|---|
| National Institute of General Medical Sciences | GM107026 | Thomas J Maresca |

The funders had no role in study design, data collection and interpretation, or the decision to submit the work for publication.

### Author contributions

Vikash Verma, Conceptualization, Resources, Data curation, Formal analysis, Investigation, Methodology, Writing—original draft, Writing—review and editing; Thomas J Maresca, Conceptualization, Data curation, Formal analysis, Supervision, Funding acquisition, Investigation, Methodology, Writing—original draft, Project administration, Writing—review and editing

### Author ORCIDs

Vikash Verma http://orcid.org/0000-0003-2371-4164
Thomas J Maresca https://orcid.org/0000-0003-2214-8674

### Decision letter and Author response

Decision letter https://doi.org/10.7554/eLife.38968.042
Author response https://doi.org/10.7554/eLife.38968.043

## Additional files

### Supplementary files

• Transparent reporting form
DOI: https://doi.org/10.7554/eLife.38968.040

## Data availability

All data generated or analysed during this study are included in the manuscript and supporting files.

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
