## [Decision Letter]

Thank you for sending your article entitled "Microtubule plus-tips act as signaling hubs for positioning the cleavage furrow during cytokinesis" for peer review at *eLife*. Your article is being evaluated by Anna Akhmanova as the Senior Editor, a guest Reviewing Editor, and two reviewers.

Both reviewers found the demonstration that centralspindlin can associate with plus tips and promote RhoA activation interesting, they both had concerns about the functional evidence indicating that CS-Tips are important for cytokinesis. However, if you are able to strengthen this evidence, the manuscript could become suitable for publication.

In particular, there are a number of experimental issues that would need to be addressed.

First, to what extent are the marker proteins overexpressed? To what extent is formation of CS-TIPs dependent on their over-expression? Can you assess the extent of overexpression of the proteins by western blotting and can you use IMF to detect CS-tips in WT cells? If they are strongly over expressed and you are not able to detect CS-Tips in wt cells, this would necessitate tagging of the endogenous copies of selected proteins to visualize CS-Tips.

Second, it is surprising that you did not see any of the previously published hallmarks of EB1 depletion in S2 cells, but it is not clear that this analysis was comprehensive. Given that the depletion is quite strong, this is surprising. Do you not detect any changes in spindle length or microtubule dynamics that recapitulate that which was seen before? If not, and your results deviate from previously published work, can all the phenotypes of the RNAi be rescued by re-expression of RNAi resistant EB1?

Third, the strongest functional evidence for CS-Tips comes from the GAP domain mutant; in particular because EB1 depletion could cause cytokinesis defects independent from blocking CS-Tip formation. However, the mutations in the GAP domain might cause additional defects beyond blocking their association with EB1. It would be imperative to rescue that mutant by tethering the protein to + TIPs by an independent means. For example, you could add an SxIP motif to that mutant (see PMID 29259096 for feasibility). (Note these experiments should be done in the absence of WT RacGAP50C).

Fourth, it would be important to evaluate the extent of the requirement for CS-TIPs in normally dividing S2 cells (see point 1 reviewer 1).

*Reviewer #1:*

Verma and Maresca use TIRF imaging of flattened, dividing S2 cells to analyze the distribution of key cytokinetic regulators. They report the visualization of MT plus tips decorated with centralspindlin and ECT-2 and associated with foci of RhoA activation. Depletion of EB1 prevents these factors from accumulating on microtubule tips and results in a modest, but significant reduction in cleavage furrow formation. The authors identify a region of RacGAP50C that is proposed to mediate its association with EB1. The authors propose that these plus tip complexes constitute a critical means by which RhoA is activated. While many aspects of these observations have been previously demonstrated (eg. PMID 19720876, refs in point 5 below), this study connects these foci to RhoA activation in Figure 2. While this is a very interesting finding, there are a number of significant weaknesses. First, it is not clear to what extent the experimental set up contributes to the requirement for EB1. Additionally, there are a number of major points that require additional documentation and quantification, additional controls. There are also some inconsistencies with the existing literature and other places where citations are missing or inappropriate.

Essential revisions:

1) While TIRF imaging provides an exceptional view of the ventral surface of the cell, it would also be important to visualize the events on the dorsal surface and the rest of the cell. In order to flatten the cells for TIRF imaging, they are adhered to ConA-coated slides. While this approach is frequently used in the context of S2 cells, this strong adherence and cell flattening may alter the requirements for contractile ring assembly and ingression (for example see PMID 27298323).

To provide context, therefore, the authors should show images of cells (+/- EB1) dividing on ConA slides in non-TIRF mode so that the extent and rate of furrow formation can be assessed and compared to cells that are not so highly flattened. The authors should also assess whether the requirement for EB1 in cell division is also observed in cells that are not adhered to ConA-coated glass (to compare to the results of Figure 6C). Likewise, to provide a positive control for potent cytokinesis defects, the authors should deplete the Cyk4 ortholog and assess cytokinesis defects as shown in Figure 6C,D.

2) There are discrepancies between the results shown here and those of Rogers et al. 2002. "EB1 depletion in *Drosophila* S2 cells has also been reported to cause spindle defects (Rogers et al., 2002), however we did not observe any obvious spindle morphology defects. Although, we reproducibly achieved {greater than or equal to} 90% EB1 depletion, this discrepancy could result from different levels of EB1 depletion."

Rogers et al. demonstrated a reduction in microtubule dynamicity, do the present authors also observe this effect? Might the inability to see spindle defects be due to the use of TIRF imaging which limits the depth of view?

3) Title: "Microtubule plus-tips act as signaling hubs for positioning the cleavage furrow during cytokinesis."

In the absence of the controls mentioned in point 1, the data shown do not compellingly demonstrate defects in cleavage furrow positioning in the absence of these signaling hubs on MT plus-tips. For example, while the 4 fold increase in non dividing cells sounds impressive, the absolute number is not particularly high (10-15%). Relatedly, how can more cells fail to form furrows than become binucleates?

4) Figure 1

How is the moment of anaphase onset determined in TIRF? What determines the time of "pre-patterning" and "post-patterning". For example, the times stated in Figure 1 vary between 1A and 1B.

5) Figure 4

The requirement for Aurora B for microtubule association of centralspindlin has been shown previously. (PMID 19962307, 20451386).

That centralspindlin can localize in an aurora B dependent manner without aurora B being normally localize has been shown previously (PMID 15263015, Figure 2)

The role of Plk1 in promoting assembly of the centralspindlin-Ect2 complex is well known. (PMID 19468302, 19468300, 17488623, 17360533)

The localization of centralspindlin to microtubule tips is reminiscent to that shown by Hu and Mitchison (PMID 18411311) in monopolar cells. This should be cited and discussed.

While the authors cite the relevant work, these figures appear superfluous.

6) Figure 6

The deletion of four amino acids in the middle of a globular region of a protein is likely to have severe consequences. The fact that the protein still localizes is not a particularly good control for this mutation, as the localization to the midzone is mediated by the N-terminus of the protein, a completely separate domain. It also seems unlikely that this peptide would be sufficiently accessible for EB1 binding. The data shown are equally consistent with this mutation disrupting an interaction with a binding partner that contains the EB1 binding motif. It is particularly risky to mutate the proline residue. To convince this reviewer otherwise, it would be necessary to show that these mutations do not impair the ability of the GAP domain to bind to RhoA•GTP. The dominant negative effect is perplexing according to the authors model as simply preventing centralspindlin tracking to MT tips would be predicted to be less severe than EB1 depletion, which would lead to delocalization of many proteins including centralspindlin. Given the centrality of this point to the manuscript, this point requires significant attention.

7) Perhaps one of the most interesting findings is the temporal shift from rather global tip accumulation to equatorially focused tips. Given the overlap of the present work with previously published work, it would be reasonable for the authors to better understand the basis for this shift.

8) There is compelling evidence that cortically associated microtubules are largely dispensable for Rho dependent contractility (PMID 10837228, 20008563). Thus, there is not a strong, general requirement for EB1-dependent centralspindlin binding to plus tips. Yet the authors state, "Altogether, these results overwhelmingly support the conclusion that CS-TIPs recruit the Rho-GEF, Dm ECT2, to activate cortical RhoA, which results in localized myosin accumulation and induction of cortical contractility."

In general, the text should describe how these plus tips may be one of several means by which cells generate centralspindlin-activated Ect2 at the plasma membrane (PMID 29738735).

9) The first two pages of the introduction rehash the astral stimulation/astral inhibition/central spindle debate. Given the compelling evidence from a variety of systems that there are multiple means by which cells can generate a region with high RhoA activity, this extensive discussion is rather superfluous.

10) Overall the manuscript is insufficiently quantitative. Figure 1, Figure 2A, Figure 3, Figure 5 all show interesting phenomena. In general, throughout the manuscript, it is not clear how many cells were observed with particular localization patterns, nor what fraction of cells in a similar cell cycle stage exhibit this pattern. The colocalization results are similarly qualitative.

11) What are the expression levels of the transgenes relative to the endogenous proteins? Can these abundant foci be a consequence of the overexpression? Can endogenous proteins be detected by IMF at the corresponding sites in the absence of overexpression, or can they be detected with fluorescent tagging of the endogenous genes?

*Reviewer #2:*

Using the *Drosophila* S2 cell line and TIRF microscopy, Verma and Maresca observed microtubule (MT) plus-end and/or cortical localization of several critical cytokinesis regulators, including two kinases, Aurora B and Polo. Abrogation of plus-end accumulation of these factors increased the frequency of cytokinesis failure, which led to the conclusion that MT plus tips act as signaling hubs for cytokinesis.

The provided video data is of high quality and convincing in terms of the dynamics of each factor under normal condition and also after kinase inhibition. Publication of this study would be warranted if the authors could address the following important issue.

Essential revisions:

I was not fully convinced of the key data of this manuscript: abrogation of MT tip localization of cytokinesis factors led to cytokinesis failure. In the current manuscript, the authors presented two data sets to support the conclusion.

1) EB1 depletion eliminated MT plus-end accumulation of the cytokinesis factors and produced binucleated cells.

2) Deletion of an EB1-binding motif abrogated MT plus-end accumulation of RacGAP50C, and this mutant protein dominantly caused cytokinesis failure, when over-expressed.

Regarding the first evidence, versatile functions are known for EB1, including the regulation of MT plus-end dynamics, and it is unclear which function(s) of EB1 causes cytokinesis defect. The data is consistent with the authors' conclusion but does not serve as a direct proof of it.

Contrarily, assessment of the RacGAP50C mutant function directly establishes the role of plus-end accumulation in cytokinesis. However, the current experimental design, which employs over-expression of the mutant in S2 cells that possess intact RacGAP50C, assuming that the mutant has a dominant-negative effect, is not sophisticated. The authors should instead perform an alternative experiment, where endogenous RacGAP5C is depleted while the mutant (RNAi-insensitive version) is ectopically and transiently expressed. In addition, to verify the authors' model, the Rho sensor and/or myosin dynamics should be analyzed in the replacement condition, rather than just counting the number of binucleated cells.

[Editors' note: the decision letter after re-review follows.]

Thank you for resubmitting your work entitled "Microtubule plus-tips act as physical signaling hubs to activate RhoA during cytokinesis" for further consideration at *eLife*. Your revised article has been favorably evaluated by Anna Akhmanova (Senior Editor) and a guest Reviewing Editor.

The manuscript has been improved but there are some remaining issues that need to be addressed before acceptance, as outlined below:

The detection of centralspindlin at tips of a subset of astral microtubules and their ability to induce RhoA activation is interesting and well documented. It is a nice complement to the observations made in bipolar and monopolar cells. It suggests that the balance between centralspindlin accumulating on individual plus ends and bundled plus ends can vary more than previously thought.

However, many of the results in Figure 3 and Figure 4 of the manuscript show a significant overlap with previous studies. It is well known that Aurora B kinase activity (CPC might be a better acronym) is required for centralspindlin's association with both MT. Likewise, it is well established that Plk1 activity is required for centralspindlin to activate ECT-2 and hence promote RhoA activity. It is also known that MKLP2 contributes to CPC localization but CPC function in centralspindlin localization is known to be MKLP2 independent. These findings have been established in different systems and reviewed numerous times including a comprehensive review from earlier this year (Basant & Glotzer, Current Biology).

Therefore, we suggest focusing the paper into a shorter report that describes the detection of CS-TIPs, their EB1 dependence, and the high spatial and temporal correlations between the colocalization of centralspindlin, ECT-2, and active RhoA. This is a really nice visualization of a series of connections that has been studied before, but with a significantly better spatiotemporal precision.

1) Most importantly, there are concerns about the functional analysis that is included in the paper. The key experiment from the revision was to address whether the cytokinesis defect caused by the depletion of EB1 resulted from the displacement of centralspindlin from MT ends. To address this question, one would want to generate an EB1-centralspindlin fusion protein (and controls) and assess the frequency of cytokinesis failures after depletion of EB1 – (i.e. does this correct the defect in Figure 6 C,D). Of course, one would need to make the EB1 part of the fusion resistant to RNAi and preferably mutate the SxIP and the PT binding motifs to ensure that only centralspindlin tip tracks under these conditions.

The experiment provided in Figure 6F-J was to ask whether Cyk4 derivatives with a fused EB1 can rescue depletion of endogenous Cyk4. The results shown do not seem to prove conclusively that this is rescuing cytokinesis by virtue of its ability to Tip Track as opposed to other means by which of centralspindlin can induce RhoA activation.

Related to this experiment, the methods section does not cover how you did the rescue experiments with Cyk4(Figure 6) or MKLP1 (Figure 1F). The section entitled "RNA interference (RNAi) experiments" does not describe Cyk4 or MKLP1 depletion methodology. The text states " Although RNAi primers (targeting 3'UTR + 200 base pairs of coding sequence) were designed to deplete the endogenous copy, we often observed a partial depletion of the transgene as well, which can be seen in all the western blots." Shouldn't those 200 bp of the rescue construct be re-encoded?

[Editors' note: further revisions were requested prior to acceptance, as described below.]

Thank you for submitting your article "Microtubule plus-tips act as physical signaling hubs to activate RhoA during cytokinesis" for consideration by *eLife*. The evaluation of your article has been overseen by a Reviewing Editor and Anna Akhmanova as the Senior Editor.

The Reviewing Editor has drafted this decision to help you prepare a revised submission.

Summary:

Verma and Maresca use TIRF imaging of flattened, dividing S2 cells to analyze the distribution of key cytokinetic regulators. They report the visualization of MT plus ends decorated with centralspindlin and ECT-2 and associated with foci of RhoA activation. Depletion of EB1 prevents these factors from accumulating on microtubule tips and results in a modest, but significant reduction in cleavage furrow formation. The modest increase in cytokinesis defects resulting from loss of EB1 can be rescued by a version of RACGAP50C that is targeted to plus tips by direct fusion to EB1. The authors identify a region of RacGAP50C that is proposed to mediate its association with EB1. The authors propose that these plus end complexes contribute to RhoA activaton during cytokinesis.

Essential revisions:

1) As mentioned in the previous correspondence, the term CS-TIP is somewhat ambiguous and this remains the case even in the revised version.

In the Introduction the term is defined as "Dm MKLP1, RacGAP50C, ABK, and Polo each localize to and track astral MT plus-tips within minutes of anaphase onset before being lost from a majority of polar astral MTs and retained on equatorial astral MTs. These specialized MT plus-tips were deemed 'cytokinesis signaling TIPs', referred to hereafter as CS-TIPs, because they recruited cortical ECT2 and locally activated RhoA."

This suggests that a CS-TIP includes centralspindlin, Polo, Aurora B, and ECT-2.

However, the kinetics of accumulation of Aurora B, ECT2 and centralspindlin are distinct. For example, in subsection “The centralspindlin complex, ABK, and Polo kinase localize to astral MT plus-tips following anaphase onset and become patterned onto equatorial astral MTs over time” "ABK localization on CS-TIPs was evident for less time (~1 minute) than the centralspindlin components, which typically lasted for 2-10 minutes on polar MTs and >10 minutes on equatorial MTs."

Subsection “CS-TIPs recruit cortical Dm ECT2 (Pebble) upon contact and activate RhoA” "Unlike the CS-TIP components, Dm ECT2 did not robustly tip-track on astral MTs (although rare events were observed), rather, cortical Dm ECT2 co-localized with MT plus-tips within 10.5 {plus minus} 1.0 seconds (mean {plus minus} SEM, n = 21) of contact (Figure 2B), which was followed by amplification of Dm ECT2 recruitment that peaked 1.5 {plus minus} 0.1 (mean {plus minus} SEM, n = 18) minutes after the plus-tips contacted the cortex (Figure 2C; Video 6)."

Thus, it remains ambiguous when a plus end becomes a CS-TIP, particularly due to the absence of systematic colocalization data. Is it when centralspindlin accumulates or Aurora B and centralspindlin, or the subset of these tips that accumulate ECT2 when they reach the cortex?

2) In subsection “CS-TIPs recruit cortical Dm ECT2 (Pebble) upon contact and activate RhoA”, the results are not sufficiently clearly indicated. When n=23 or n=18 >=95% is either 100% or <95%. Also, the statistics are written as n=22; 20/22 cells. Please use a single format. number/total cells is the simplest.

3) In subsection “Investigating the contribution of midzone components (ABK, Polo kinase, kinesin-4, MKLP2) to CS-TIP assembly”, the manuscript extensively describes PRC1. The experiment that follows assesses the role of Klp3A. While Klp3A can function with PRC1, PRC1 can function without it, so this section should focus more on Klp3A and less on PRC1, which is not studied.

4) Subsection “Satellite MT arrays analogous to midzone and astral MTs assemble in proximity to the cortex” "compelling MT structures" Compelling is highly subjective, "MT structures that assembled." would be more accurate.

5) Subsection “EB1 is required for robust MT plus-tip localization of CS-TIP components and contributes to furrow initiation and successful completion of cytokinesis” and following. "A four fold increase, relative to control cells" is unclear, as the level in control cells is not stated in the text. Please include the raw number for depleted and control cells somewhere in the text of this section.

6) Subsection “RacGAP50C mutants fused to EB1 support MT plus-tip based activation of RhoA and can rescue the cytokinesis defects of RacGAP50C depletion” does not seem to add much to the manuscript. First, there is the unexpected complexity that the ∆PTIV variant acts as a dominant negative, increasing the basal frequency of binucleate cells, which is unexplained. Second, as this construct can function both at MT tips and off MT tips, it does not allow for any additional conclusions.

7) While von Dassow, 2009 is discussed in subsection “CS-TIP- vs midzone-based signaling in positioning the cleavage furrow”, the results of that paper directly contradict the speculation also in subsection “CS-TIP- vs midzone-based signaling in positioning the cleavage furrow”.

"Another interesting feature of the CS-TIP mode of signaling is that it could play a more critical role in larger cells such as those in early embryonic divisions where the midzone MT array is far from the cortex." von Dassow, 2009 clearly shows that astral tips are dispensable for cytokinetic positioning in large blastomeres. This is the reason why the reference was mentioned in previous rounds.

8) Subsection “CS-TIP- vs midzone-based signaling in positioning the cleavage furrow”. It would be appropriate to cite Basant and Glotzer, 2018, where this explanation was previously proposed.

9) Subsection “Role of EB1 and a conserved hxxPTxh motif in RacGAP50C in cytokinesis”. Please report the "reported differences" from McKinley and Cheeseman, 2017 and Yang, 2017.

10) The title: the expression "plus-tips", although sometimes used in the literature, is somewhat ambiguous, as it is similar to the term "+TIPs", which refers to microtubule plus end tracking proteins. Please consider rewording the title.

---

## [Author Response]

Reviewer #1:

Verma and Maresca use TIRF imaging of flattened, dividing S2 cells to analyze the distribution of key cytokinetic regulators. They report the visualization of MT plus tips decorated with centralspindlin and ECT-2 and associated with foci of RhoA activation. Depletion of EB1 prevents these factors from accumulating on microtubule tips and results in a modest, but significant reduction in cleavage furrow formation. The authors identify a region of RacGAP50C that is proposed to mediate its association with EB1. The authors propose that these plus tip complexes constitute a critical means by which RhoA is activated. While many aspects of these observations have been previously demonstrated (eg. PMID 19720876, refs in point 5 below), this study connects these foci to RhoA activation in Figure 2. While this is a very interesting finding, there are a number of significant weaknesses. First, it is not clear to what extent the experimental set up contributes to the requirement for EB1. Additionally, there are a number of major points that require additional documentation and quantification, additional controls. There are also some inconsistencies with the existing literature and other places where citations are missing or inappropriate.Essential revisions:1) While TIRF imaging provides an exceptional view of the ventral surface of the cell, it would also be important to visualize the events on the dorsal surface and the rest of the cell. In order to flatten the cells for TIRF imaging, they are adhered to ConA-coated slides. While this approach is frequently used in the context of S2 cells, this strong adherence and cell flattening may alter the requirements for contractile ring assembly and ingression (for example see PMID 27298323).To provide context, therefore, the authors should show images of cells (+/- EB1) dividing on ConA slides in non-TIRF mode so that the extent and rate of furrow formation can be assessed and compared to cells that are not so highly flattened. The authors should also assess whether the requirement for EB1 in cell division is also observed in cells that are not adhered to ConA-coated glass (to compare to the results of Figure 6C). Likewise, to provide a positive control for potent cytokinesis defects, the authors should deplete the Cyk4 ortholog and assess cytokinesis defects as shown in Figure 6C,D.

Thank you for raising this point. To address this reviewer’s concern, we have now performed several experiments and modified the manuscript accordingly. Each of these points are addressed below and labelled A, B, C, and D.

A) To observe the tip localization towards the dorsal surface and the rest of the cell, we have now performed spinning disk confocal microscopy of the cells expressing *Dm* MKLP1-EGFP and Tag-RFP-αtubulin, including cells where endogenous *Dm* MKLP1 has been depleted (see point 11 below). Our spinning disk confocal imaging of *Dm* MKLP1-EGFP reveled plus tip localization not only on the ventral side of the cell close to the coverslip, but also towards the dorsal side (up to 4.8µm) and throughout the cell volume. We have now included a new Figure 1G, and a spinning disk confocal Z-section movie (Video 5) in the manuscript.

B) To address the concern regarding the use of ConA to flatten S2 cells, we have now included a new video (Video 1) and text to clarify this issue (see subsection “The centralspindlin complex, ABK, and Polo kinase localize to astral MT plus-tips following anaphase onset and become patterned onto equatorial astral MTs over time” in the manuscript). We would like to point out that we have not observed any defects in contractile ring assembly and constriction when S2 cells are adhered to ConA. The figure below is adapted from videos we previously published (Ye, Torabi and Maresca, 2016) from spinning disk confocal time-lapse imaging of a dividing S2 cell coexpressing mCherry-α-tubulin and MRLC-EGFP. 3-D reconstructions of the confocal Z-stacks show myosin ring assembly and restriction on ConA-coated cover-glass. To address this concern in the present study, we conducted overnight time-lapse imaging of S2 cells seeded on ConA and found that 97% of dividing cells ingress furrows within ~10 minutes of anaphase onset. Of these cells 32% complete cytokinesis normally while 68% of cells regress their furrows, but after 39.4 ± 7.6 minutes. Thus, the ConA concentration we use on our cover-glass appears to negatively affect cytokinetic abscission or some other aspect of the latest stages of cytokinesis, but does not affect the establishment of the cleavage furrow or furrow ingression, which is the process we are studying in this work.

C) We would also like to note that the percent multinucleated cells reported for +/- EB1 conditions are reflecting the behaviors of semi-adherent S2 cells that were growing in typical plastic-bottomed 6-well tissue culture plates for 4-days and not flattened on ConA until they were seeded for ~45 minutes before the cells were fixed and stained.

D) To provide the positive control for the cytokinesis defects, we have now depleted RacGAP50C in various cell lines (8 times), and, we observed a ~20-32-fold increase in the percent of binucleate cells over the baseline cells measured in control RNAi conditions (See Figure 6J; Figure 1—figure supplement 1 H, I and Figure 6—figure supplement 2F).

2) There are discrepancies between the results shown here and those of Rogers et al. 2002. "EB1 depletion in Drosophila S2 cells has also been reported to cause spindle defects (Rogers et al., 2002), however we did not observe any obvious spindle morphology defects. Although, we reproducibly achieved {greater than or equal to} 90% EB1 depletion, this discrepancy could result from different levels of EB1 depletion."Rogers et al., demonstrated a reduction in microtubule dynamicity, do the present authors also observe this effect? Might the inability to see spindle defects be due to the use of TIRF imaging which limits the depth of view?

Thank you for pointing this out. As you rightly noted, we did not “comprehensively” analyze other aspects (aside from CS-TIP assembly and cytokinesis defects) of the depletion phenotype. This comment led us to revisit our data by measuring spindle lengths in control and EB1-depleted cells. Consistent with Rogers et al., 2002, we did indeed measure a reduction in spindle length in EB1-depleted cells compared to control cells (control RNAi: 7.0 +/- 0.86 µm, EB1 RNAi: 6.1 +/- 0.69 µm; two-tailed p-value from student’s t-test = 4.31 E-12). The 13% reduction in spindle length that we measured may differ from the 30% reduction measured in Rogers et al., due to differences in experimental conditions such as live-cell (us) versus fixed-cell (Rogers) analyses and/or duration of the RNAi treatments (4 day (us) vs. 7 day (Rogers)).

Nonetheless, we now feel confident that our EB1 depletion conditions are recapitulating changes in spindle length consistent with published EB1 findings. We have now included new text (subsection “EB1 is required for robust MT plus-tip localization of CS-TIP components and contributes to furrow initiation and successful completion of cytokinesis”) and a supplementary figure (Figure 6—figure supplement 1B) in the manuscript.

3) Title: "Microtubule plus-tips act as signaling hubs for positioning the cleavage furrow during cytokinesis."In the absence of the controls mentioned in point 1, the data shown do not compellingly demonstrate defects in cleavage furrow positioning in the absence of these signaling hubs on MT plus-tips. For example, while the 4 fold increase in non dividing cells sounds impressive, the absolute number is not particularly high (10-15%). Relatedly, how can more cells fail to form furrows than become binucleates?

Thank you for raising this point. We feel that we have adequately addressed your concerns raised in point 1, but your comments led us to revisit our Title. We do feel that CS-TIPs are required for RhoA activation and, therefore, we have now changed the title to reflect this. Our Title now reads ‘’Microtubule plus-tips act as physical signaling hubs to activate RhoA during cytokinesis’’.

Given the importance of cytokinesis, we feel that 3-4-fold increase in cytokinesis failure will be problematic for the cells. However, we recognize that this lower number compared to the depletion of MKLP1 or RacGAP50C could be indicative of redundant pathways that require centralspindlin complex. We have now modified the text in various places to mention the likely importance of redundant pathways.

With respect to your comment regarding the differences in furrow initiation failure and binucleates, we would like to point out that these data sets were obtained from two different experiments. Live-cell microscopy was performed to evaluate the extent of furrow initiation defects, while IF on fixed cells was performed to evaluate the changes in binucleate formation, so some variation in the data sets could result from these differences.

4) Figure 1How is the moment of anaphase onset determined in TIRF? What determines the time of "pre-patterning" and "post-patterning". For example, the times stated in Figure 1 vary between 1A and 1B.

In our experiments, α-tubulin is fluorescently tagged, so anaphase onset in the TIRF field is determined by observing the elongation of spindle microtubules. Additionally, because of the accompanying changes in the MT dynamics and distribution more microtubules start to appear in the TIRF filed during anaphase onset. This was also observed and reported in the Vale et al., 2009.

We defined pre-patterning as the time point when uniform localization of CS-TIP components (MKLP1, RacGAP50C, ABK, and Polo kinase) is visible on astral MT plus tips (both polar and equatorial), while post-patterning is defined as the time point when the vast majority of polar CS-TIPs are lost and are retained on the equatorial MTs. Not surprisingly, there is cell-to-cell variability in this phenomenon. In fact, some cells retain a few stray polar CS-TIPs for more than 10 minutes. We have now included this text in the figure legend of Figure 1.

5) Figure 4The requirement for Aurora B for microtubule association of centralspindlin has been shown previously. (PMID 19962307, 20451386).That centralspindlin can localize in an aurora B dependent manner without aurora B being normally localize has been shown previously (PMID 15263015, Figure 2)The role of Plk1 in promoting assembly of the centralspindlin-Ect2 complex is well known. (PMID 19468302, 19468300, 17488623, 17360533)The localization of centralspindlin to microtubule tips is reminiscent to that shown by Hu and Mitchison (PMID 18411311) in monopolar cells. This should be cited and discussed.While the authors cite the relevant work, these figures appear superfluous.

In response to A, B, and C, we agree with reviewers that requirement of ABK in centralspindlin clustering and localization at the midzone has been shown previously and discussed extensively. However, these studies (PMID 19962307, 20451386) did not address the contribution of ABK activity to MT plus-tip association of the centralspindlin complex. We feel that we have advanced the previous knowledge of cleavage furrow establishment in at least two ways: (1) We have provided high resolution live imaging data for centralspindlin complex, ECT2, and Rhotekin (active RhoA) dynamics in *Drosophila* S2 cells that previous reports lack, and further we have shown that RhoA can be activated by CS-TIP based signal, which we believe is considerable addition to the existing knowledge. (2) Most prior studies have concentrated on midzone-based signaling; our data is presented and analyzed in the context of an interesting and under-studied population at the MT +TIPs. We feel that it was important to establish what activities are important for TIP-based localization in comparison to established requirements for midzone localization. Similarly, requirement of Plk1 in ECT2 recruitment is well known, as this reviewer pointed out, but none of those studies focused on microtubule +TIP-based association of Plk1, therefore our study is important to establish the relative contribution of ABK and Polo in CS-TIPs signaling.

Thank you for pointing out the Hu and Mitchison comparison! We completely agree with your point, and this is indeed a great addition. We have now expanded this point in Discussion section.

6) Figure 6The deletion of four amino acids in the middle of a globular region of a protein is likely to have severe consequences. The fact that the protein still localizes is not a particularly good control for this mutation, as the localization to the midzone is mediated by the N-terminus of the protein, a completely separate domain. It also seems unlikely that this peptide would be sufficiently accessible for EB1 binding. The data shown are equally consistent with this mutation disrupting an interaction with a binding partner that contains the EB1 binding motif. It is particularly risky to mutate the proline residue. To convince this reviewer otherwise, it would be necessary to show that these mutations do not impair the ability of the GAP domain to bind to RhoA•GTP. The dominant negative effect is perplexing according to the authors model as simply preventing centralspindlin tracking to MT tips would be predicted to be less severe than EB1 depletion, which would lead to delocalization of many proteins including centralspindlin. Given the centrality of this point to the manuscript, this point requires significant attention.

We agree with this reviewer that deleting a proline residue could be problematic, however, to evaluate the functional significance this conserve motif (PTIV), there are not that many alternative methods available, therefore, we proceeded to delete this motif. To address this reviewers concern and following the editor’s advice as to where to focus our experimental efforts, we have now restored the tip tracking abilities of RacGAP mutants (∆C and ∆PT) via fusing them to EB1 (Figure 6I,J). While we were unable to make stable cell cells with RacGAP mutants, fusion of EB1 allowed us make stable cell lines in which at least 50% of cells were expressing the GFP-tagged transfected construct. This allowed us to conduct functional rescue experiments in the absence of endogenous RacGAP50C protein using WT fused to EB1 as a control. The new findings are described in subsection ‘’RacGAP50C mutants fused to EB1 support MT plus-tip based activation of RhoA and can rescue the cytokinesis defects of RacGAP50C depletion‘’ and a new figures (Figure 6J; Figure 1—figure supplement 1H, I and Figure 6—figure supplement 2F) is added to accommodate this change. Since we observed a comparable rescue of the% binucleates in RacGAP50C depleted cells for both WT and δ PT mutants, we infer that aside from tip localizing abilities other cytokinesis-related functionalities of the ∆PTIV mutant are preserved. However, with reference to your specific point about the GAP activity, it has been shown in *Drosophila* that the GAP activity is dispensable for cytokinesis (Goldstein et al., 2005). Further, you raised a point about the accessibility issue of the PTIV motif for EB1 binding, in the revised manuscript, we have now raised this point and discuss it in significantly greater detail (see subsection “Role of EB1 and a conserved hxxPTxh motif in RacGAP50C in cytokinesis”) with relationship to recent work we have published on the motor protein NOD.

7) Perhaps one of the most interesting findings is the temporal shift from rather global tip accumulation to equatorially focused tips. Given the overlap of the present work with previously published work, it would be reasonable for the authors to better understand the basis for this shift.

We fully agree with the reviewer that this is indeed an interesting finding, however, we feel that this line of investigation falls beyond the scope of the present study. This will certainly be a future line of investigation.

8) There is compelling evidence that cortically associated microtubules are largely dispensable for Rho dependent contractility (PMID 10837228, 20008563). Thus, there is not a strong, general requirement for EB1-dependent centralspindlin binding to plus tips. Yet the authors state, "Altogether, these results overwhelmingly support the conclusion that CS-TIPs recruit the Rho-GEF, Dm ECT2, to activate cortical RhoA, which results in localized myosin accumulation and induction of cortical contractility."In general, the text should describe how these plus tips may be one of several means by which cells generate centralspindlin-activated Ect2 at the plasma membrane (PMID 29738735).

Thank you for pointing this out. We recognized that redundancy exists, and we have now changed the manuscript to reflect these points (see our Discussion section).

9) The first two pages of the introduction rehash the astral stimulation/astral inhibition/central spindle debate. Given the compelling evidence from a variety of systems that there are multiple means by which cells can generate a region with high RhoA activity, this extensive discussion is rather superfluous.

We feel that the relevant information is necessary to provide the context/background information, and it is especially helpful for the readers who are not familiar with the background, so we would like to keep the Introduction as such.

10) Overall the manuscript is insufficiently quantitative. Figure 1, Figure 2A, Figure 3, Figure 5 all show interesting phenomena. In general, throughout the manuscript, it is not clear how many cells were observed with particular localization patterns, nor what fraction of cells in a similar cell cycle stage exhibit this pattern. The colocalization results are similarly qualitative.

Thank you for pointing this out. We have now included the n value and the relevant information regarding the quantitation in the figure legends and in the text when pertinent to avoid any confusion.

11) What are the expression levels of the transgenes relative to the endogenous proteins? Can these abundant foci be a consequence of the overexpression? Can endogenous proteins be detected by IMF at the corresponding sites in the absence of overexpression, or can they be detected with fluorescent tagging of the endogenous genes?

Thank you for pointing this out, we feel that we have now done significant work to address this issue which is shown in Figure 1F and Figure 1—figure supplement 1E.

To address this issue, we have performed western blots of S2 cell lines expressing centralspindlin complex. Quantitation of western blots indicates that expression of Pavarotti-GFP transgene is 5-fold lower than the endogenous Pavarotti (see Figure 1F). Similarly, RacGAP50C transgene expression is lower than its endogenous counterpart, and is barely detectable by RacGAP50C antibody. Since our proteins are not overexpressed, we decided not to perform the IF experiments. Moreover, previous studies in *Drosophila*andmammalian cells as well as *Xenopus* embryos have reported MT plus-tip localization of MKLP1 and MgcRacGAP (centralspindlin complex) as well as the CPC component ABK (Breznau et al., 2017; Nishimura and Yonemura, 2006; Vale et al., 2009).

Reviewer #2:

Using the Drosophila S2 cell line and TIRF microscopy, Verma and Maresca observed microtubule (MT) plus-end and/or cortical localization of several critical cytokinesis regulators, including two kinases, Aurora B and Polo. Abrogation of plus-end accumulation of these factors increased the frequency of cytokinesis failure, which led to the conclusion that MT plus tips act as signaling hubs for cytokinesis.The provided video data is of high quality and convincing in terms of the dynamics of each factor under normal condition and also after kinase inhibition. Publication of this study would be warranted if the authors could address the following important issue.Essential revisions:I was not fully convinced of the key data of this manuscript: abrogation of MT tip localization of cytokinesis factors led to cytokinesis failure. In the current manuscript, the authors presented two data sets to support the conclusion.1) EB1 depletion eliminated MT plus-end accumulation of the cytokinesis factors and produced binucleated cells.2) Deletion of an EB1-binding motif abrogated MT plus-end accumulation of RacGAP50C, and this mutant protein dominantly caused cytokinesis failure, when over-expressed.Regarding the first evidence, versatile functions are known for EB1, including the regulation of MT plus-end dynamics, and it is unclear which function(s) of EB1 causes cytokinesis defect. The data is consistent with the authors' conclusion but does not serve as a direct proof of it.Contrarily, assessment of the RacGAP50C mutant function directly establishes the role of plus-end accumulation in cytokinesis. However, the current experimental design, which employs over-expression of the mutant in S2 cells that possess intact RacGAP50C, assuming that the mutant has a dominant-negative effect, is not sophisticated. The authors should instead perform an alternative experiment, where endogenous RacGAP5C is depleted while the mutant (RNAi-insensitive version) is ectopically and transiently expressed. In addition, to verify the authors' model, the Rho sensor and/or myosin dynamics should be analyzed in the replacement condition, rather than just counting the number of binucleated cells.

Thank you for pointing this out. These points (1 and 2) were also raised by reviewer 1 and the monitoring editor, so we have put significant efforts to address these issues. To avoid redundancy, please see our response to point 6 above.

Following your advice, we now show that fusion of EB1 to ∆C-RacGAP50C mutant restored its tip tracking activity and triggered activation and focusing of RhoA near CS-TIPs (see Figure 6—figure supplement 2B) when endogenous RacGAP50C was depleted from cells. We have now discussed this observation in detail in the subsection “RacGAP50C mutants fused to EB1 support MT plus-tip based activation of RhoA and can rescue the cytokinesis defects of RacGAP50C depletion” of the manuscript.

[Editors' note: the decision letter after re-review follows.]

The manuscript has been improved but there are some remaining issues that need to be addressed before acceptance, as outlined below:The detection of centralspindlin at tips of a subset of astral microtubules and their ability to induce RhoA activation is interesting and well documented. It is a nice complement to the observations made in bipolar and monopolar cells. It suggests that the balance between centralspindlin accumulating on individual plus ends and bundled plus ends can vary more than previously thought.However, many of the results in Figure 3 and Figure 4 of the manuscript show a significant overlap with previous studies. It is well known that Aurora B kinase activity (CPC might be a better acronym) is required for centralspindlin's association with both MT. Likewise, it is well established that Plk1 activity is required for centralspindlin to activate ECT-2 and hence promote RhoA activity. It is also known that MKLP2 contributes to CPC localization but CPC function in centralspindlin localization is known to be MKLP2 independent. These findings have been established in different systems and reviewed numerous times including a comprehensive review from earlier this year (Basant & Glotzer, Current Biology).Therefore, we suggest focusing the paper into a shorter report that describes the detection of CS-TIPs, their EB1 dependence, and the high spatial and temporal correlations between the colocalization of centralspindlin, ECT-2, and active RhoA. This is a really nice visualization of a series of connections that has been studied before, but with a significantly better spatiotemporal precision.

In response to your point, we have removed the old Figure 4, which showed that inhibition of ABK and Polo kinase activities affected RhoA activation and ECT2 recruitment. We have kept the video files that show the wash-in moves while imaging Rhotekin and ECT2 and reference them in the text. We also maintained Figure 3 because we feel it is important to describe the contribution of these various factors to CS-TIP assembly. This is especially important for the ABK inhibition as a prior publication (Vale, Spudich and Griffis, 2009), which used RNAi, concluded that ABK was not required for the plus-tip localization of *Dm* MKLP1.

1) Most importantly, there are concerns about the functional analysis that is included in the paper. The key experiment from the revision was to address whether the cytokinesis defect caused by the depletion of EB1 resulted from the displacement of centralspindlin from MT ends. To address this question, one would want to generate an EB1-centralspindlin fusion protein (and controls) and assess the frequency of cytokinesis failures after depletion of EB1 – (i.e. does this correct the defect in Figure 6 C,D). Of course, one would need to make the EB1 part of the fusion resistant to RNAi and preferably mutate the SxIP and the PT binding motifs to ensure that only centralspindlin tip tracks under these conditions.The experiment provided in 6 F-J was to ask whether Cyk4 derivatives with a fused EB1 can rescue depletion of endogenous Cyk4. The results shown do not seem to prove conclusively that this is rescuing cytokinesis by virtue of its ability to Tip Track as opposed to other means by which of centralspindlin can induce RhoA activation.Related to this experiment, the methods section does not cover how you did the rescue experiments with Cyk4(Figure 6) or MKLP1 (Figure 1F). The section entitled "RNA interference (RNAi) experiments" does not describe Cyk4 or MKLP1 depletion methodology. The text states " Although RNAi primers (targeting 3'UTR + 200 base pairs of coding sequence) were designed to deplete the endogenous copy, we often observed a partial depletion of the transgene as well, which can be seen in all the western blots." Shouldn't those 200 bp of the rescue construct be re-encoded?

We conducted the EB1-RacGAP50C hybrid experiment you suggested and found that, contrary to EB1 depletion from WT cells, there was not an increase in the percent of binucleates when endogenous EB1 was depleted from cells expressing EB1-RacGAP50C.

[Editors' note: further revisions were requested prior to acceptance, as described below.]

Essential revisions:1) As mentioned in the previous correspondence, the term CS-TIP is somewhat ambiguous and this remains the case even in the revised version.In the Introduction the term is defined as "Dm MKLP1, RacGAP50C, ABK, and Polo each localize to and track astral MT plus-tips within minutes of anaphase onset before being lost from a majority of polar astral MTs and retained on equatorial astral MTs. These specialized MT plus-tips were deemed 'cytokinesis signaling TIPs', referred to hereafter as CS-TIPs, because they recruited cortical ECT2 and locally activated RhoA."This suggests that a CS-TIP includes centralspindlin, Polo, Aurora B, and ECT-2.However, the kinetics of accumulation of Aurora B, ECT2 and centralspindlin are distinct. For example, in subsection “The centralspindlin complex, ABK, and Polo kinase localize to astral MT plus-tips following anaphase onset and become patterned onto equatorial astral MTs over time” "ABK localization on CS-TIPs was evident for less time (~1 minute) than the centralspindlin components, which typically lasted for 2-10 minutes on polar MTs and >10 minutes on equatorial MTs."Subsection “CS-TIPs recruit cortical Dm ECT2 (Pebble) upon contact and activate RhoA” "Unlike the CS-TIP components, Dm ECT2 did not robustly tip-track on astral MTs (although rare events were observed), rather, cortical Dm ECT2 co-localized with MT plus-tips within 10.5 {plus minus} 1.0 seconds (mean {plus minus} SEM, n = 21) of contact (Figure 2B), which was followed by amplification of Dm ECT2 recruitment that peaked 1.5 {plus minus} 0.1 (mean {plus minus} SEM, n = 18) minutes after the plus-tips contacted the cortex (Figure 2C; Video 6)."Thus, it remains ambiguous when a plus end becomes a CS-TIP, particularly due to the absence of systematic colocalization data. Is it when centralspindlin accumulates or Aurora B and centralspindlin, or the subset of these tips that accumulate ECT2 when they reach the cortex?

The specific examples mentioned above have been changed. The manuscript has also been edited to more specifically define when we view that a MT plus-end is capable of acting as Cytokinesis Signaling (CS) tip and a statement as to the relevant components for signaling has also been added.

2) In subsection “CS-TIPs recruit cortical Dm ECT2 (Pebble) upon contact and activate RhoA”, the results are not sufficiently clearly indicated. When n=23 or n=18 >=95% is either 100% or <95%. Also, the statistics are written as n=22; 20/22 cells. Please use a single format. number/total cells is the simplest.

The consistent designation of number/total has been made throughout the manuscript.

3) In subsection “Investigating the contribution of midzone components (ABK, Polo kinase, kinesin-4, MKLP2) to CS-TIP assembly”, the manuscript extensively describes PRC1. The experiment that follows assesses the role of Klp3A. While Klp3A can function with PRC1, PRC1 can function without it, so this section should focus more on Klp3A and less on PRC1, which is not studied.

A discussion of Klp3A to this section has been added.

4) Subsection “Satellite MT arrays analogous to midzone and astral MTs assemble in proximity to the cortex” "compelling MT structures" Compelling is highly subjective, "MT structures that assembled." would be more accurate.

The change has been made.

5) Subsection “EB1 is required for robust MT plus-tip localization of CS-TIP components and contributes to furrow initiation and successful completion of cytokinesis” and following. "A four fold increase, relative to control cells" is unclear, as the level in control cells is not stated in the text. Please include the raw number for depleted and control cells somewhere in the text of this section.

The specific percent binucleates for depleted and controls has been added to this section.

6) Subsection “RacGAP50C mutants fused to EB1 support MT plus-tip based activation of RhoA and can rescue the cytokinesis defects of RacGAP50C depletion” does not seem to add much to the manuscript. First, there is the unexpected complexity that the ∆PTIV variant acts as a dominant negative, increasing the basal frequency of binucleate cells, which is unexplained. Second, as this construct can function both at MT tips and off MT tips, it does not allow for any additional conclusions.

The section (and associated data from the Figures) has been removed. The Figures have been edited accordingly.

7) While von Dassow, 2009 is discussed in subsection “CS-TIP- vs midzone-based signaling in positioning the cleavage furrow”, the results of that paper directly contradict the speculation also in subsection “CS-TIP- vs midzone-based signaling in positioning the cleavage furrow”."Another interesting feature of the CS-TIP mode of signaling is that it could play a more critical role in larger cells such as those in early embryonic divisions where the midzone MT array is far from the cortex." von Dassow, 2009 clearly shows that astral tips are dispensable for cytokinetic positioning in large blastomeres. This is the reason why the reference was mentioned in previous rounds.

The sentence has been removed.

8) Subsection “CS-TIP- vs midzone-based signaling in positioning the cleavage furrow”. It would be appropriate to cite Basant and Glotzer, 2018, where this explanation was previously proposed.

The citation to Basant and Glotzer has been added.

9) Subsection “Role of EB1 and a conserved hxxPTxh motif in RacGAP50C in cytokinesis”. Please report the "reported differences" from McKinley and Cheeseman, 2017 and Yang, 2017.

McKinley et al., reported spindle abnormalities in ~̴ 35% mitotic cells upon double knockout of EB1 and EB3; however, Yang et al. reported seemingly normal mitotic spindles for the cells where all three EB proteins (EB1, EB2, and EB3) were knocked out.

10) The title: the expression "plus-tips", although sometimes used in the literature, is somewhat ambiguous, as it is similar to the term "+TIPs", which refers to microtubule plus end tracking proteins. Please consider rewording the title.

The term in the title has been changed from “plus-tips” to “plus-ends”.